# Source attribution of Arctic black carbon and sulfate aerosols and associated Arctic surface warming during 1980–2018

Lili Ren[1], Yang Yang[1*], Hailong Wang[2], Rudong Zhang[2], Pinya Wang[1], Hong Liao[1]

[1]Jiangsu Key Laboratory of Atmospheric Environment Monitoring and Pollution Control, Jiangsu Collaborative Innovation Center of Atmospheric Environment and Equipment Technology, School of Environmental Science and Engineering, Nanjing University of Information Science and Technology, Nanjing, Jiangsu, China

[2]Atmospheric Sciences and Global Change Division, Pacific Northwest National Laboratory, Richland, Washington, USA

*Correspondence to yang.yang@nuist.edu.cn

**Abstract**

Observations show that the concentrations of Arctic sulfate and black carbon (BC) aerosols have declined since the early 1980s. Previous studies have reported that reducing sulfate aerosols potentially contributed to the recent rapid Arctic warming. In this study, a global aerosol-climate model equipped with an Explicit Aerosol Source Tagging (CAM5-EAST) is applied to quantify the source apportionment of aerosols in the Arctic from sixteen source regions and the role of aerosol variations in affecting changes in the Arctic surface temperature from 1980 to 2018. The CAM5-EAST simulated surface concentrations of sulfate and BC in the Arctic had a decrease of 43% and 23%, respectively, in 2014–2018 relative to 1980–1984, mainly due to the reduction of emissions from Europe, Russia and Arctic local sources. Increases in emissions from South and East Asia led to positive trends of Arctic sulfate and BC in the upper troposphere. All aerosol radiative impacts are considered including aerosol-radiation and aerosol-cloud interactions, as well as black carbon deposition on snow and ice-covered surfaces. Within the Arctic, sulfate reductions caused a top-of-the-atmosphere (TOA) warming of 0.11 and 0.25 W m$^{-2}$, respectively, through aerosol-radiation and aerosol-cloud interactions. While the changes in Arctic atmospheric BC has little impact on local radiative forcing, the decrease of BC in snow/ice led to a net cooling of 0.05 W m$^{-2}$. By applying climate sensitivity factors for different latitudinal bands, global changes in sulfate and BC during 2014–2018 (with respect to 1980–1984) exerted a +0.088 K and 0.057 K Arctic surface warming, respectively, through aerosol-radiation interactions. Through aerosol-cloud interactions, the sulfate reduction gave an

Arctic warming of +0.193 K between the two time periods. The weakened BC effect on snow/ice albedo led to an Arctic surface cooling of –0.041 K. The changes in atmospheric sulfate and BC outside the Arctic totally produced an Arctic warming of +0.25 K, the majority of which is due to the mid-latitude changes in radiative forcing. Our results suggest that changes in aerosols over the mid-latitudes of the Northern Hemisphere have a larger impact on Arctic temperature than other regions through enhanced poleward heat transport. The combined total effects of sulfate and BC produced an Arctic surface warming of +0.297 K, explaining approximately 20% of the observed Arctic warming since the early 1980s.

## 1. Introduction

The Arctic has warmed rapidly since the 1980s, with a 1.5 K increase in the surface air temperature, which is about two to four times faster than the global average (Trenberth et al., 2007; Serreze et al., 2009). The significant rise in air and ground temperatures occurred in phase with dramatic melting of Arctic sea ice and snow, potentially contributing to Arctic amplification (Pithan and Mauritsen, 2014; Zhang et al., 2019). A number of studies have examined possible mechanisms that caused the rapid Arctic warming (Graversen et al., 2008; Screen and Ian, 2010; Screen and Simmonds, 2010; Alexeev et al., 2012; Zhang et al., 2018). Observations and modeling studies suggest that, although anthropogenic long-lived greenhouse gases (GHGs) dominate the radiative forcing of the climate system, variations in black carbon (BC) aerosol and other short-lived air pollutants are a good explanation for the faster Arctic warming (Law and Andreas, 2007; Quinn et al., 2008; Shindell et al., 2008). In particular, Shindell and Faluvegi (2009) found that aerosols may have warmed the Arctic surface during 1976-2010 based on model sensitivity experiments. The aerosols that caused Arctic warming are not only from local emissions. Studies have shown that changes in long-range transport of sulfate and BC aerosols from mid-latitude regions have caused strong wintertime warming in the Arctic (e.g., Breider et al., 2014; Fisher et al., 2011; Shindell et al., 2008). In addition, the mid-latitude aerosols can influence Arctic climate through changing poleward heat transport (Navarro et al., 2016).

Observed and modeled seasonal cycles of aerosol concentrations at the remote Arctic surface show a maximum in winter, a phenomenon commonly known as Arctic

Haze, and a minimum in summer (Law and Andreas, 2007; Quinn et al., 2007; Eckhardt
et al., 2015; Garrett et al., 2010; Sharma et al., 2006). The winter maximum has been
attributed to the long-range transport of anthropogenic pollution from the mid-latitudes
of the Northern Hemisphere and weak removal in the Arctic (Stohl, 2006; Wang et al.,
2014). In contrast, summer aerosol concentrations in the Arctic atmosphere reach a
minimum value due to a reduced poleward aerosol transport from the mid-latitudes and
efficient wet scavenging processes during the transport (Bourgeois and Bey, 2011;
Browse et al., 2012; Garrett et al., 2011). Anthropogenic aerosol species (e.g., sulfate,
BC and organic matter) can affect Arctic climate by disturbing the energy balance of
the earth system (Yang et al., 2019a). Sulfate aerosols directly scatter solar radiation
and indirectly influence cloud processes by serving as cloud condensation nuclei (Yang
et al., 2017a; Zamora et al., 2017; Zhao and Garrett, 2015). BC absorbs solar radiation
and warms the atmosphere (Bond et al., 2013; Yang et al., 2017b; Lou et al., 2019a),
which can increase or decrease cloud cover depending on the vertical distribution of
BC relative to clouds (e.g., McFarquhar and Wang, 2006; Lou et al., 2019b). When it
deposits on snow and ice, BC can reduce surface albedo and accelerate snow melt
(Flanner et al., 2007; Qian et al., 2015). Breider et al. (2017) estimated the aerosol
radiative forcing due to aerosol-radiation interactions in the Arctic and found that,
averaged over 2005–2010, the top-of-the-atmosphere (TOA) forcing is -0.60 $\pm$
0.02 $Wm^{-2}$ for sulfate and +0.44 $\pm$ 0.04 $Wm^{-2}$ for BC over the Arctic.
Analysis of long-term changes in sulfate and BC can help to gain a comprehensive
understanding of their past and present impacts on the Arctic climate. In situ

observations of sulfate and BC concentrations in the Arctic (e.g., at Alert, Barrow, Station Nord, and Zeppelin) have shown a declining trend since the 1980s (Gong et al., 2010; Heidam et al., 1999; Hirdman et al., 2010; Quinn et al., 2009; Sharma et al., 2004; Sharma et al., 2006; Sinha et al., 2017; Sirois and Barrie, 1999). Based on the chemical transport model (GEOS-Chem) simulations, Breider et al. (2017) found that annual sulfate and BC concentrations decreased by 2–3% per year over the Arctic during 1980-2010. McConnell et al. (2007) presented a historical BC trend derived from ice-core records, showing that BC concentration had been declining steadily after the peak around 1910.

Source attribution analysis of atmospheric aerosols in the Arctic, which can help to understand aerosol trends, is extremely important for air pollution research. There is less local anthropogenic aerosol emission in the Arctic region than in polluted regions of the world. Pollutants in the Arctic are generally from mid-latitude areas by long-distance transport (Fisher et al., 2011; Wang et al., 2014). Recent studies have found that Arctic aerosols mainly originated from Eurasia, Southeast Asia, Siberia and North America (Fisher et al., 2011; Qi et al., 2017; Sharma et al., 2013; Stohl, 2006). The contribution of Eurasia to Arctic sulfate and BC aerosols concentration is dominant in the lower atmosphere, while South and Central Asia contributed the most at high altitudes (e.g., Wang et al., 2014). In general, Northern Europe and Russia, with large industrial emissions, are the main source region of Arctic BC aerosols in spring (Rahn et al., 1977; Rahn, 1981; Raatz and Shaw, 1984; Barrie, 1986; Koch and Hansen, 2005; Sharma et al., 2006; Stohl, 2006). Shindell et al. (2008) studied the sensitivity of

simulated Arctic aerosol concentrations to emissions perturbations in 2001 and found that European emissions contributed to Arctic sulfate concentrations near the surface and at 500 hPa by 73% and 51%, respectively. East Asia has the largest contribution at 250 hPa, reaching 36%. Based on simulations of a chemical transport model, Fisher et al. (2011) concluded that West Asia emissions dominated wintertime Arctic sulfate concentration, with contributions between 30% and 45%. In the past few decades, anthropogenic emissions have changed rapidly, with a decrease in Europe and North America and an increase in South and East Asia. This may have had an important impact on the Arctic aerosols and climate (Breider et al., 2014).

In this study, the global aerosol-climate model CAM5 (Community Atmosphere Model, version 5) equipped with an Explicit Aerosol Source Tagging (CAM5-EAST) is used to examine the attribution of Arctic aerosols to 16 different source regions and the aerosol-related Arctic warming during 1980–2018. We focus on changes in sulfate and BC near-surface concentrations, total column burden, and radiative forcing as well as their impacts on the surface temperature over the Arctic. Sulfate and BC concentrations from the CAM5-EAST model and observations at remote Arctic stations are compared. CAM5-EAST tagging results are used to quantify the contributions of different sources to the decadal changes in Arctic sulfate and BC surface concentrations and vertical profiles. Based on the Arctic climate sensitivity factors, we estimate the responses of the Arctic surface temperature to the variations in sulfate and BC during the analyzed time periods.

## 2. Methodology

## 2.1 Model Description and Experimental Setup

The global aerosol-climate model CAM5, which is the atmospheric component of the earth system model CESM (Community Earth System Model, Hurrell et al., 2013) developed at the National Center for Atmospheric Research (NCAR), is used to simulate Arctic aerosols and climate for years 1980–2018 (after one-year model spin-up). In this model version, mass and number concentrations of sulfate particles are predicted for the three lognormal modes (i.e., Aitken, accumulation, and coarse modes) of the three-mode modal aerosol module (Liu et al., 2012) in CAM5. Aerosols are internally mixed in the same aerosol mode and then externally mixed between modes. Within each mode, sulfate is internally mixed with primary/secondary organic matter, BC, mineral dust, and/or sea salt. BC is mixed with other aerosol species (e.g., sulfate, POA, SOA, sea salt, and dust) in the accumulation mode immediately after being emitted into the atmosphere without considering explicit aging processes. The optical properties and radiative impact of aerosols are calculated online. The model also includes climate effects of aerosols through aerosol-radiation and aerosol-cloud interactions.

In this study, the model is configured to run at a horizontal grid of 1.9° latitude × 2.5° longitude with 30 vertical layers up to 3.6 hPa. The CAM5 simulation is conducted with prescribed time-varying solar radiation, sea surface temperature, sea ice concentration, GHGs, and emissions of aerosols and their precursor gases. Sea surface temperatures and sea ice concentrations are created from the merged Reynolds/HADISST products, as described in Hurrell et al. (2008). Solar radiation and

GHGs follow the CMIP6 configuration for AMIP-type of simulations. In order to better
reproduce the aerosol transport driven by large-scale circulations in the model, the wind
field is nudged toward the MERRA-2 (Modern Era Retrospective-Analysis for
Research and Applications, Version 2) reanalysis (Rienecker et al., 2011; Gelaro et al.,
2017) at a 6-hourly relaxation timscale.
**2.2 Explicit Aerosol Source Tagging and Source Regions**
The EAST was implemented in CAM5 to quantify the source-receptor relationships
of aerosols in recent studies (Wang et al., 2014;Yang et al., 2017a,b; 2018a,b,c). All
physical, chemical and dynamical processes of aerosols for each tagged source region
or sector are considered independently and consistently by using additional sets of
aerosol variables in CAM5-EAST, which is different from the widely used emission
sensitivity method that assumes a linear response to emission perturbation or the
indirect method of tracing long-lived constituents associated with particular sources.
Without such assumption of linear response or constant decaying rate, EAST is more
physically accurate than the source attribution methods mentioned above. In this study,
sulfate and BC are explicitly tracked throughout the processes from source emissions
to deposition in a single model simulation.
We focus on the Arctic (66.5°N–90°N) as the receptor region in this study. According
to source region definition of the Hemispheric Transport of Air Pollution model
experiment phase 2 (HTAP2), sulfate and BC from 16 regions are tagged (Fig. 1):
Europe (EUR), North America (NAM), Central America (CAM), South America
(SAM), North Africa (NAF), South Africa (SAF), the Middle East (MDE), Southeast
Asia (SEA), Central Asia (CAS), South Asia (SAS), East Asia (EAS), Russia-Belarus-
Ukraine (RBU, hereafter Russia), Pacific-Australia-New Zealand (PAN), the Arctic
(ARC), Antarctic (ANT), and Non-Arctic/Antarctic Ocean (OCN). Note that the OCN
tag includes sources from oceans and volcanic eruptions.
## 2.3 Radiative Forcings and Temperature Response
Radiative forcing (RF) due to aerosol-radiation interactions is calculated as the
difference of clear-sky net radiative fluxes at TOA between two separate diagnostic
calculations, including and excluding a specific aerosol in the radiative transfer
calculation, respectively (Ghan et al., 2012). Aerosols interact with stratiform clouds
through two-moment microphysics, in which the nucleation of stratiform cloud droplets
is based on the scheme of Abdul-Razzak and Ghan et al. (2000). Although aerosols have
no microphysical impact on convective clouds, the ambient temperature and convection
can be affected by BC-induced atmospheric heating. The Arctic equilibrium
temperature response is estimated using Arctic climate sensitivity factors ($\lambda$, K W$^{-1}$m$^2$),
defined as the change in Arctic surface temperature per unit RF for different latitudinal
bands from Sand et al. (2016) and Shindell and Faluvegi (2009). The change in
equilibrium temperature response is defined as $\Delta T = \sum_{j=LAT} \lambda_j * \Delta RF_j$. $\Delta$ represents
the difference of the annual mean of a variable for a specific year compared to the
average during 1980–1984 in this study. RF is radiative forcing due to aerosol-radiation
or aerosol-cloud interactions associated with sulfate or black carbon. LAT represents
latitudinal bands over the Arctic (60°N–90°N), Northern Hemisphere mid-latitudes
(28°N–60°N), tropics (28°S–28°N) and Southern Hemisphere (90°S–28°S). Many
studies used these climate sensitivity factors to estimate the Arctic temperature
responses using RF calculated from different models (e.g., Sand et al., 2016). However,
we note that, since the λ values were calculated with a different climate model (NASA-
GISS), the estimated Arctic equilibrium temperature response based on these factors
could be biased.

### 2.4 Aerosol and Precursor Emissions

In order to simulate the long-term temporal variations in aerosols, historical
anthropogenic (Hoesly et al., 2018) and biomass combustion (van Marle et al., 2017)
emissions of aerosols and precursor gases during 1980–2014 are used in the simulation
following the CMIP6 (Coupled Model Intercomparison Project Phase 6) protocol. For
the most recent years (2015-2018), yearly interpolated emissions from the SSP (Shared
Socioeconomic Pathways) 2-4.5 scenario are used, which is the modest scenario
compared to other SSPs and is widely utilized in many MIPs (O'Neill et al., 2016).
Figure 1 and Figure 2 (Figure S1) show the spatial distribution and time series of annual
anthropogenic $SO_2$ and BC emissions, respectively, during 1980–2018, from the 16
source regions. The global total anthropogenic $SO_2$ and BC emission rates, averaged
over 1980–2018, are 118.4 Tg yr$^{-1}$ and 8.1 Tg yr$^{-1}$, respectively. $SO_2$ emissions are
relatively high in East Asia (23.6 Tg yr$^{-1}$), Europe (15.8 Tg yr$^{-1}$) and North America
(15.4 Tg yr$^{-1}$), while BC emissions show high values in East Asia (1.8 Tg yr$^{-1}$), South
Africa (1.6 Tg yr$^{-1}$) and South Asia (0.9 Tg yr$^{-1}$). Comparing 2014–2018 to 1980–1984,
global anthropogenic $SO_2$ emission was reduced by 32.2 Tg yr$^{-1}$ (24.8% relative to
1980–1984). The largest decreases took place in Europe (83.0%), North America
(80.7%) and Russia (74.8%). In East Asia, the emission of anthropogenic $SO_2$ increased
by a factor of 2.7 from 1980 to 2014, followed by a decreasing trend after 2014 due to
stricter air pollution regulations. The global anthropogenic BC emission increased from
6.5 Tg yr$^{-1}$ in 1980 to a peak of 9.6 Tg yr$^{-1}$ in 2014, followed by a slow decline, with an
overall increase of 42% between the first and last five years of 1980–2018. Regionally,
compared to 1980–1984, averaged BC emissions in 2014–2018 in Europe and Russia
decreased by 45.2% and 44.1%, respectively, while BC emissions in East Asia and
South Asia almost increased by a factor of 2. Within the Arctic, $SO_2$ and BC emissions
decreased by 5.8% and 38.3%, respectively.
**2.5 Model Evaluation**
To assess the ability of the model to simulate Arctic sulfate and BC, Figs. 3 and 4
compare simulated near-surface concentrations of sulfate and BC, respectively, in
spring and summer during 1980–2018 with observations at five Arctic stations: Alert
(82°N, 62°W), Station Nord (81°N, 16°W), Barrow (71°N, 156°W), Ny-Alesund (78°N,
11°E) and Kevo (69°N, 27°E). The observations are derived from European Monitoring
and Evaluation Programme and World Data Centre for Aerosols database
(http://ebas.nilu.no) and Breider et al. (2017).
Overall, the sulfate and BC concentrations in spring is higher than those in summer,
mainly due to lower removal rate and more efficient transport (Stohl, 2006). According
to previous CAM5 studies on aerosol wet removal and long-range transport, the model
underestimates aerosol concentrations in spring, likely due to biases in
parameterizations of convective transport and wet scavenging of aerosols (Bond et al.,
2013; Liu et al., 2011; Wang et al., 2013; Qian et al., 2014; Yang et al., 2018a). All sites
show that sulfate concentrations decrease during the analyzed time period and BC
decreases at specific sites, which can be explained by the reduction of non-local
emissions as illustrated by the source attribution. Compared to the observed values, the
model can reasonably simulate the time variations of sulfate and BC in the Arctic but
the magnitude at some of the sites is largely underestimated. The Kevo site (69°N,
27°E), which is close to Western Eurasia, is the only site that has both sulfate and BC
data for more than 30 years. At this site, the simulated sulfate in spring and summer
decreased at a rate of -3.18% and -1.92% per year, respectively, which are similar to -
4.37% and -3.26% per year from observations. The decreasing rates of BC in spring
and summer were -2.89% and -1.74%, respectively, that are also consistent with the
observed values of -3.01% and -2.82%.
Observational data are very limited in the Arctic, especially the long-term
observations. The available BC measurements are equivalent elemental carbon (EBC),
which is usually obtained by converting the light absorbed by the particles accumulated
on the ground instrument filter into the BC concentration. The uncertainty in optical
properties of BC makes this conversion challenging. Other light absorbing substances,
such as dust and organic carbon, also affect the BC measurements, so EBC would tend
to be higher than the actual BC concentration. Researchers found that BC observations
could be biased by 30% to 200% (Sharma et al., 2017; Sinha et al., 2017) due to the
inclusion of other light absorption components in the atmosphere. Shindell et al. (2008)
and Koch et al. (2009) found great differences between the current models and
observations of Arctic BC and sulfate through multi-model comparation studies,
including incorrect seasonality and order of magnitude biases. Given the large apparent
discrepancies in BC for all models, it is difficult to determine the relative authenticity
of the models using currently available data (Shindell et al., 2008).

## 3. Source Apportionment of Aerosols in the Arctic

The near-surface concentrations of sulfate and BC over the Arctic can be
quantitatively attributed to both Arctic local emissions and remote sources outside the
Arctic through the source tagging in CAM5-EAST. The absolute and relative source
contributions of emissions from the major source regions to the simulated annual mean
near-surface sulfate and BC concentrations averaged over the Arctic (66.5°N–90°N) are
shown in Fig. 5 and Fig. S2. Arctic local emissions and sources near the Arctic (e.g.,
Europe and Russia) are the main contributors to the near-surface concentrations of
Arctic sulfate and BC. Relative to the average of 0.447 $\mu g/m^3$ during 1980–1984, the
simulated annual sulfate concentration over the Arctic has a decrease of 42.8% (0.191
$\mu g/m^3$) in 2014–2018 (Table 1). Sulfate concentration shows a considerable decreasing
trend from 1980 to 2000, which then slows down after 2000. The decrease in sulfate
during this time period primarily results from the reduction in emissions from Europe
and Russia, which contributes to 18.6% (0.083 $\mu g/m^3$) and 18.8% (0.084 $\mu g/m^3$) of the
decline of the Arctic sulfate concentrations, respectively. The change in emissions from
Central Asia and North America, respectively, explains 1.6% (0.007 $\mu g/m^3$) and 3.4%
(0.015 $\mu g/m^3$) of the reduced concentration.
Simulated Arctic BC concentration also shows a considerable decline before 2000,
but a slight rise after 2000, which is consistent with the BC observations at Alert.
Overall, the average concentration of BC in the Arctic had a decrease of 22.98% (3.7
ng/m$^3$ relative to the 1980–1984 average of 16.1 ng/m$^3$) in 2014–2018, mainly due to
the reductions in emissions originating from the Arctic and Russia, which lead to 9.32%
(1.5 ng/m$^3$) and 14.91% (2.4 ng/m$^3$) of the decrease (Table 1). Sources in Europe, North
America, and East Asia account for less than 4% of the changes in Arctic near-surface
BC concentration. The remaining source regions (Central America, South America,
North Africa, South Africa, the Middle East, Southeast Asia, Central Asia, South Asia,
Pacific-Australia-New Zealand, Antarctic, and Non-Arctic/Antarctic Ocean) have no
substantial impact on the BC concentration in the Arctic (total contribution less than
2%) due to the weak emission strength or long transport pathways. Since the Arctic
sulfate and BC aerosol concentrations contributed by non-local sources have been
reducing, the fractional contribution of Arctic local source increased from 33.6% and
53.4% to 55.1% and 57.3%, respectively. To further reduce present-day or future
aerosols in the Arctic, efforts can be made to control local sources in the Arctic as well
as emissions from Russia. The industry and energy sectors account for the majority of
local sources in the Arctic (Fig. S4). Reducing the emissions of industry and energy
sectors may be effective for the reduction of local sulfate and BC concentrations in the
Arctic.
Aerosols are often transported across continents in the free troposphere rather than
near the surface, resulting in a higher relative contribution of non-local sources to the
aerosol concentration at higher altitudes than near the surface. Figure 6 shows the
vertical profiles of absolute and relative contributions of major source regions to sulfate
and BC concentrations in the Arctic. Different source regions have very distinct vertical
distributions of their contributions. Below 1 km, Arctic local emissions account for the
majority of Arctic sulfate and BC concentrations. For BC and sulfate located between
1 km and 5 km, emissions from Russia are the major sources. Above 8 km, East Asia
and South Asia are the major source regions of the Arctic aerosol concentrations, which
is consistent with results using other models (e.g., Shindell et al., 2008). Arctic and
Russia have their maximum absolute contributions at 0.2 km and 1.4 km, respectively.
Europe and North America have their maximum absolute contributions around 2 km.
The contribution of East Asia and South Asia increases with the increase of altitude,
reaching their maximum contribution values at 8 km and 11 km, respectively. Previous
studies also pointed out that, in April, BC showed a high concentration in the mid-
troposphere of the Arctic, mainly due to the effect of Asian anthropogenic aerosols, that
are transported to the Arctic through warm conveyor belt (Wang et al., 2011). Evidence
from aircraft and ground-based measurements showed that eastern and southern Asia
source regions contributed the most to the BC concentration in the Arctic mid-
troposphere, while northern Asia dominated the contribution to the Arctic surface BC
(Abbatt et al., 2019).
The changes in source contributions to the annual mean vertical profile of sulfate and
BC concentrations over the Arctic between 2014–2018 and 1980–1984 are shown in
Fig. 7. Below 6 km, due to the effective emission reduction, the contribution from
Europe and Russia to the Arctic sulfate was each decreased by nearly 0.1 $\mu$g m$^{-3}$ in
2014–2018, compared to 1980–1984. North America contribution also had a slight
decline below 2 km. Between 10–15 km, contributions from South Asia and East Asia
increased at the upper troposphere, which is consistent with the increase in emissions
over these regions, leading to a combined increase in sulfate concentration of up to 0.1
$\mu g\ m^{-3}$ at the upper troposphere of the Arctic. The BC concentration below 2 km
contributed by Arctic and Russia emissions each had a decrease of up to 2 ng $m^{-3}$, which
dominated the decrease of BC concentration in the Arctic lower atmosphere. Similar to
sulfate, BC concentrations contributed by East Asia and South Asia increased in the
high altitudes (Breider et al. 2017, Fisher et al., 2011; Qi et al., 2017; Sharma et al.,
2013; Stohl, 2006), mainly due to increased emissions in these two regions, offsetting
the decrease in column burden owing to the reduced loading in the lower atmosphere.
Similar to our findings, Breider et al. (2017) found that the simulated decrease in
aerosol optical depth in the Arctic from 1980 to 2010 was driven by a strong decrease
in aerosol loading at lower altitudes due to the emission changes in West Eurasia, Russia
and North America and an increase in aerosols at higher altitudes resulting from the
changes in emissions in regions such as South Asia and East Asia.
A linear regression approach is applied in order to analyze the trends of the annual
near-surface concentrations and column burden of sulfate and BC from 1980 to 2018
shown in Fig. 8 and the individual source contributions to these trends are summarized
in Table 2. During 1980–2018, simulated Arctic near-surface concentration and column
burden of sulfate decreased by 20% and 13 % per decade, respectively. Due to the air
pollution regulations in Europe and dissolution of the former Soviet Union, reductions
in emissions from Europe and Russia led to decreasing trends of 7–10% per decade in
the near-surface concentration and column burden of sulfate, having the largest
contributions to sulfate trends among all tagged source regions. In addition, the change
in North America emissions contributed to a 2–4% per decade decreasing trend in the
Arctic sulfate concentration and burden, which is related to its emission control since
1980s. South and East Asia together contributed to an increase of total Arctic sulfate
burden at a rate of 8% per decade, associated with the emission rise during this time
period. The near-surface concentration of Arctic BC has a decreasing trend of 12% per
decade during 1980–2018, mostly driven by the decreases in contributions from Russia
and Arctic local emissions (6% per decade each). For BC column burden, the decreasing
trends contributed by the reductions in emissions from Russia and Europe are offset by
the increasing trends caused by emission increases in South and East Asia, resulting in
an insignificant change of total BC burden during 1980–2018. All trend values
mentioned above are statistically significant at the 95% confidence level.
## 4. Aerosol Radiative Forcing and Associated Arctic Warming
Both sulfate and BC influence the Arctic climate through perturbing atmospheric and
surface radiation balance. The spatial distribution of the climatological mean TOA
radiative forcing due to aerosol-radiation interactions ($RF_{ari}$) of sulfate and BC averaged
over 1980–2018 is shown in Fig. 9. The Arctic sulfate exerts a negative $RF_{ari}$ primarily
by scattering incoming solar radiation back into the space, with the forcing in range of
-0.4~0 $Wm^{-2}$. The atmospheric BC can absorb solar radiation in the atmosphere and
leads to a positive $RF_{ari}$ of 0.1~0.4 $Wm^{-2}$ in the Arctic, which is similar to the values of
$0.1\sim0.6$ Wm$^{-2}$ estimated in previous studies (Koch and Hansen, 2005; Flanner et al.,
2009; AMAP, 2011; Bond et al., 2011; Samset et al., 2014; Wang et al., 2014). In the
high and mid-latitudes of the Northern Hemisphere, the RF$_{ari}$ of sulfate over Europe
and Russia is in the range of $-1.0\sim-0.4$ Wm$^{-2}$. Sulfate RF$_{ari}$ over North America varies
from $-0.2$ Wm$^{-2}$ to $-1.0$ Wm$^{-2}$. The negative RF$_{ari}$ of sulfate over East Asia is more than
$-1.0$ Wm$^{-2}$, mainly due to the high sulfate concentrations. BC over Europe, Russia and
Central Asia exerts a positive RF$_{ari}$ of $0.4\sim1$ Wm$^{-2}$. The BC RF$_{ari}$ over East Asia reaches
a high value over $1.0$ Wm$^{-2}$.

Previous studies have suggested that Arctic climate responds not only to Arctic local

forcings but also to forcings outside the Arctic due to the meridional energy transport
change (Navarro et al., 2016). To estimate the relative roles of regional aerosol trends
in affecting the Arctic warming, we looked into the temporal variation of annual mean
radiative forcing of sulfate and BC in different latitudinal bands during 1980–2018 (Fig.
10). Within the Arctic (60°N–90°N), the magnitude of sulfate RF$_{ari}$ decreases from -
$0.21$ Wm$^{-2}$ in 1980–1984 to $-0.10$ Wm$^{-2}$ in 2014–2018, indicating a warming effect in
the Arctic from the local sulfate change. Over the mid-latitudes (28°N–60°N), the
sulfate RF$_{ari}$ decreases from $-0.87$ Wm$^{-2}$ to $-0.53$ Wm$^{-2}$ between the first and last five
years of 1980–2018, while the magnitude of the sulfate RF$_{ari}$ in the tropical region
(28°S–28°N) increases from $-0.52$ Wm$^{-2}$ to $-0.60$ Wm$^{-2}$. The positive BC RF$_{ari}$ increases
from $0.55$ Wm$^{-2}$ to $0.74$ Wm$^{-2}$ in the mid-latitudes and from $0.51$ Wm$^{-2}$ to $0.76$ Wm$^{-2}$
in the tropics, while the BC RF$_{ari}$ over the Arctic has no obvious change during this time
period.
Systematic assessment of the impact of aerosols on Arctic warming since 1980s
requires quantifying the Arctic temperature responses to changes in radiative forcing of
different aerosol species over different regions. Here, we apply Arctic climate
sensitivity factors, defined as the Arctic temperature response per unit radiative forcing,
for each short-lived climate forcers over the Arctic, mid-latitudes of the Northern
Hemisphere, tropics and Southern Hemisphere from Sand et al. (2016) and Shindell
and Faluvegi (2009) to calculate the recent Arctic surface temperature change related
to the variations in sulfate and BC radiative forcings over the different latitudinal bands
during 1980–2018 (Fig. 11 and Table 3). This method has been widely adopted to
examine the Arctic temperature response to aerosol forcings (e.g., Breider et al., 2017;
Flanner, 2013; Sand et al., 2016; Shindell and Faluvegi, 2009; Yang et al., 2018c).
It is estimated that, between 1980–1984 and 2014–2018, changes in total $RF_{ari}$ of
sulfate and BC produce a surface warming of +0.145 K over the Arctic, with +0.088 K
(61%) contributed by the sulfate forcing change and the remaining explained by the BC
forcing change. The sulfate-related Arctic warming is mainly due to the decrease in
sulfate in mid-latitudes that enhances the temperature gradient between the mid-
latitudes and Arctic, resulting in a strengthened meridional heat transport and, therefore,
the Arctic warming of +0.059 K. The change in Arctic local $RF_{ari}$ of sulfate provides
+0.035 K of the surface warming, while the forcing change in the tropics has a
negligible influence on the Arctic temperature change. The Arctic temperature
responses to increases in BC $RF_{ari}$ over the mid-latitudes and tropics are +0.029 K and
+0.031 K, respectively, related to the enhanced poleward heat transport from the
warming radiative impact in the mid-latitudes, while changes in the Arctic BC $RF_{ari}$
only exert a weak cooling of -0.005 K. Overall, the $RF_{ari}$ change over the mid-latitudes
provides the strongest warming effect (+0.088K) to the Arctic compared to other
latitude bands, owing to the aerosol-induced increase in the poleward heat transport.

While the results above focus on the effects of aerosol-radiation interactions, the

aerosol-cloud interactions ($RF_{aci}$) and BC snow/ice albedo effects can also influence
Arctic climate. Sulfate $RF_{aci}$ is estimated by scaling sulfate $RF_{ari}$ based on the ratio of
sulfate $RF_{aci}$ and $RF_{ari}$ over different latitudes from Sand et al. (2016). Within the Arctic,
the magnitude of negative TOA $RF_{aci}$ of sulfate decreases from -0.48 $Wm^{-2}$ in 1980–
1984 to -0.23 $Wm^{-2}$ in 2014–2018, indicating a warming effect due to the local sulfate
change. Over the mid-latitudes, the sulfate $RF_{aci}$ decreases from -2.46 $Wm^{-2}$ to -1.49
$Wm^{-2}$ between the first and last five years of 1980–2018, while the magnitude of the
sulfate $RF_{aci}$ in the tropical region increases from -1.78 $Wm^{-2}$ to -2.08 $Wm^{-2}$. The
positive RF due to BC in snow/ice decreases from 0.34 $Wm^{-2}$ in 1980–1984 to 0.29
$Wm^{-2}$ in 2014–2018 over the Arctic, while that over the mid-latitudes increases from
0.19 $Wm^{-2}$ to 0.23 $Wm^{-2}$.

Based on the Arctic climate sensitivities, impacts of changes in radiative forcing due

to aerosol-cloud interactions of sulfate are also estimated. The sulfate $RF_{aci}$ provides an
Arctic warming of +0.193 K between 1980–1984 and 2014–2018, with +0.165 K
contributed by the $RF_{aci}$ change over the mid-latitudes and +0.078 K resulting from the
Arctic $RF_{aci}$ change. It should be noted that aerosol-cloud interactions at high latitude
regions are complicated and highly uncertain in climate models. The temperature
changes presented here only provide a rough estimate. BC in snow/ice reduces surface
albedo and increases snow/ice melt (Flanner et al., 2007; Qian et al., 2015). Due to the
decrease in Arctic BC concentration and depostion, BC concentration in the Arcitc
snow has been decreasing (e.g., Zhang et al., 2019). The weakened BC snow/ice albedo
effect leads to an Arctic cooling of $-0.061$ K, while the mid-latitude BC in snow/ice
causes an Arctic warming of $+0.019$ K. The total BC snow/ice albedo effects result in
an Arctic surface temperature change of $-0.041$ K during 1980–2018, partially
offsetting the solar absorbing effect of BC in the atmosphere. Combining all the effects,
we estimate that between 1980 and 2018, sulfate and BC contribute a total of $+0.297$ K
to the Arctic surface temperature change, approximately 20% of the observed Arctic
warming during this period.

## 5. Conclusions and Discussion

The Arctic has warmed rapidly since the 1980s, with the surface air temperature
increasing by 1.5 K. Different from the emission perturbation method that was often
used in previous studies, in this study, the EAST was implemented in CAM5 to quantify
the source attribution of aerosols in the Arctic and the aerosol-related Arctic warming
during 1980–2018. The model can reasonably simulate the spatial distribution and
temporal variation of the Arctic near-surface sulfate and BC concentrations compared
with several site measurements. Considering that the model underestimates the
magnitude of sulfate and BC concentrations, the estimated impact on Arctic
temperature from sulfate and BC could be even larger if the model were able to
accurately reproduce the measurements in the Arctic.
Compared to 1980–1984, the simulated annual average of sulfate and BC
concentrations over the Arctic in 2014–2018 had a decrease of 42.8% and 23.0%,
respectively. The decrease in emissions from Europe and Russia contributed 18.6% and
18.8% of the near-surface sulfate concentration decrease (out of 42.8%) and the
reduction in Arctic local emissions and emission from Russia led to 9.3% and 14.9% of
the BC concentration reduction (out of 23.0%), respectively. In 2014–2018, increases
in emissions from South and East Asia together contributed to an increase of sulfate
and BC concentrations up to 0.1 $\mu$g m$^{-3}$ and 2 ng m$^{-3}$, respectively, at the upper
troposphere, compared to the annual mean concentrations during 1980–1984. The
contribution of Europe and Russia emissions to the Arctic sulfate concentration each
had a decrease of about 0.1 $\mu$g m$^{-3}$ under 6 km. Below 2 km, the BC concentration
contributed by emissions from Arctic and Russia each had a decrease of up to 2 ng m$^{-}$
$^{3}$. Simulated sulfate near-surface concentration and column burden had a decreasing
trend of 20% per decade and 13% per decade, respectively, in the Arctic during 1980–
2018, mainly driven by the reductions in emissions from Europe and Russia, both of
which led to decreasing trends at a rate of 7–10% per decade. Due to the decreases in
contributions from Russia and Arctic local emissions (6% per decade each), the near-
surface concentration of Arctic BC presents a decreasing trend of 12% per decade
during 1980–2018.
Aerosols within and outside the Arctic can influence the Arctic climate through
changing the radiative balance. The magnitude of negative TOA $RF_{ari}$ of sulfate over
the Arctic decreased from -0.21 Wm$^{-2}$ in 1980–1984 to -0.10 Wm$^{-2}$ in 2014–2018. Over
the mid-latitudes, the sulfate $RF_{ari}$ magnitude decreased from -0.87 $Wm^{-2}$ to -0.53 $Wm^{-2}$
$^{2}$, while the sulfate $RF_{ari}$ over the tropics increased from -0.52 $Wm^{-2}$ to -0.60 $Wm^{-2}$. The
positive BC $RF_{ari}$ in the mid-latitudes and tropics increased from 0.55 $Wm^{-2}$ and 0.51
$Wm^{-2}$ to 0.74 $Wm^{-2}$ and 0.76 $Wm^{-2}$, respectively, while that over the Arctic had no
significant change during this time period. By applying Arctic climate sensitivity
factors obtained from the literature to the variations in aerosol radiative forcing, the
aerosol-induced Arctic surface temperature change is estimated in this study. During
1980–2018, through aerosol-radiation interactions, sulfate and BC together produced a
+0.145 K warming to the Arctic, +0.088 K (61%) of which is contributed by sulfate.
The decrease in sulfate in mid-latitudes led to an increase in Arctic temperature of
+0.059 K, whereas the Arctic local sulfate provided +0.035 K of the surface warming.
The Arctic temperature responses to changes in atmospheric BC over the mid-latitudes
and tropics are +0.029 K and +0.031 K, respectively, while changes BC in the Arctic
atmosphere only exert a weak cooling of -0.005 K. Through aerosol-cloud interactions,
sulfate exerted an Arctic warming of +0.193 K during 1980–2018, with +0.165 K
contributed by the forcing change over the mid-latitudes and +0.078 K due to the
forcing change over the Arctic. Therefore, changes in aerosols over the mid-latitudes
had the largest impact on the Arctic temperature than other regions during 1980–2018
through enhancing meridional temperature gradient and therefore poleward heat
transport, followed by changes in Arctic local aerosol forcings. Due to the decrease in
Arctic BC concentration, the weakened BC snow/ice albedo effect led to an Arctic
cooling of –0.061 K, partially offset by Arctic warming of +0.019 K induced by the BC
snow/ice albedo effect over the mid-latitudes. Combining all aerosol effects, sulfate and
BC together produced a total of +0.297 K in the Arctic surface temperature change
during 1980–2018, explaining approximately 20% of the observed Arctic warming
during this period.
Many studies have examined possible mechanisms that can explain the recent Arctic
warming, but the quantitative importance of these mechanisms is still on debate (e.g.,
Breider et al., 2017; Navarro et al. 2016). Among these mechanisms, some are related
to roles of aerosols in changing the Arctic temperature. Shindell and Faluvegi (2009)
found that aerosols may have warmed the Arctic surface due to emission reductions
during 1976-2010. Breider et al. (2017) estimated that emission reductions in
anthropogenic aerosols during 1980–2010 had contributed to a net warming at the
Arctic surface by +0.27 ± 0.04 K using the GEOS-Chem model, which is consistent
with our results. However, they did not take into consideration of the radiative forcing
from aerosol-cloud interactions and deposition of BC to snow and ice surfaces. Navarro
et al. (2016) presented simulations with an Earth system model and showed that the
reduction in European $SO_2$ emission over 1980–2005 has caused an Arctic warming by
0.5 K on annual average as a result of the enhanced poleward heat transport, which is
larger than our estimates likely due to different emissions and models used here and in
Navarro et al. (2016).There are a few sources of uncertainty in the results presented in
this study. As discussed above, the model underestimates the near-surface sulfate and
BC concentrations over the Arctic, probably due to an overly aerosol wet removal
during the long-range transport (e.g., Wang et al., 2013), uncertainties in aerosol
emissions, and biases in observations. Previous studies have reported large
discrepancies of aerosol and precursors emissions in China between MEIC (Multi-
resolution Emission Inventory for China) and CMIP6 emission inventories (e.g., Paulot
et al., 2018). The CMIP6 emissions dataset shows similar decreasing trends in
anthropogenic $SO_2$ and BC emissions over China since 2011 as in the MEIC inventory
(Fig. S3). However, the decrease of CMIP6 anthropogenic $SO_2$ and BC emissions by
39% and 0.5%, respectively, in 2017 compared to 2010 is less than the corresponding
magnitude of 62% and 27% in MEIC (Zheng et al., 2018). It indicates that the increase
in aerosol contribution from East Asia during the recent decade and its impact on Arctic
surface temperature could be overestimated in this study. Here we only discussed the
effects of sulfate and BC on the Arctic surface temperature without considering other
aerosol species, due to large uncertainties in the simulation of second organic aerosols
and the lack of other aerosol treatments (e.g., nitrate) in current model version. These
may lead to biases of the aerosol climate effects in this study. For more accurate
estimation of the aerosol-related Arctic warming, the coupled model configuration with
free running simulations should be conducted in the future. The $RF_{ari}$ calculation
follows Ghan et al. (2012), which falls into the definition of effective $RF_{ari}$ ($ERF_{ari}$),
while the climate sensitivity factors were calculated based on the stratospherically
adjusted radiative forcing. Considering that the assessment for adjusted $RF_{ari}$ ($-0.35 \pm$
$0.5$ W m$^{-2}$) is slightly lower than that for $ERF_{ari}$ ($-0.45 \pm 0.5$ W m$^{-2}$) (Boucher et al.,
2013), the temperature response could be relatively smaller than estimated here. The
relatively low model resolution may not capture the complexity of the Arctic terrain
(Yang et al., 2018c), which also induces uncertainties to the simulated aerosols in the
Arctic. High resolution or regionally refined model is more desirable if resources allow.
Given that assumed injection heights of anthropogenic emissions in models are
uncertain, the ability to simulated surface aerosol concentrations and vertical
distribution in models could also be compromised (Yang et al., 2019b). In this study,
we did not discuss the effects of meteorological parameters on the long-term aerosol
simulation mainly because the decadal aerosol variation is dominated by changes in
anthropogenic emissions rather than meteorology (Yang et al., 2016).

**Data availability.**

The CAM5 model is available at http://www.cesm.ucar.edu/models/cesm1.2/ (last access: 8 December 2019). Our CAM5-EAST model code and results can be made available through the National Energy Research Scientific Computing Center (NERSC) servers upon request. The observations are derived from European Monitoring and Evaluation Programme and World Data Centre for Aerosols database (http://ebas.nilu.no) and Breider et al. (2017).

**Competing interests.**

The authors declare that they have no conflict of interest.

**Author contribution**.

YY and HW designed the research; YY performed the model simulations; LR analyzed the data. All the authors discussed the results and wrote the paper.

**Acknowledgments.**

This research was support by the National Natural Science Foundation of China under grant 41975159, Jiangsu Specially Appointed Professor Project and the U.S. Department of Energy (DOE), Office of Science, Biological and Environmental Research as part of the Regional and Global Model Analysis program. The Pacific Northwest National Laboratory is operated for DOE by Battelle Memorial Institute

under contract DE-AC05-76RLO1830. The National Energy Research Scientific

Computing Center (NERSC) provided computational support.

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

**Table 1.** Contributions of emissions from major source regions to the simulated
annual mean near-surface sulfate and BC concentrations ($\mu$g m$^{-3}$) averaged over the
Arctic in 1980–1984 and 2014–2018, as well as the percentage differences (%)
between 1980–1984 and 2014–2018 relative to 1980–1984.

| | Sulfate Conc. | | |
|---|---|---|---|
| | **1980-1984** | **2014-2018** | **Last 5 -First 5** |
| Sum | 0.447 | 0.256 | -42.83% |
| ARC | 0.15 | 0.141 | -2.02% |
| EUR | 0.097 | 0.014 | -18.61% |
| NAM | 0.022 | 0.007 | -3.36% |
| CAS | 0.013 | 0.006 | -1.57% |
| RBU | 0.129 | 0.045 | -18.83% |
| OCN | 0.029 | 0.032 | 0.67% |
| OTH | 0.006 | 0.01 | 0.90% |
| | BC Conc. | | |
| | **1980-1984** | **2014-2018** | **Last 5 -First 5** |
| Sum | 0.0161 | 0.0124 | -22.98% |
| ARC | 0.0086 | 0.0071 | -9.32% |
| EUR | 0.0011 | 0.0006 | -3.11% |
| NAM | 0.0004 | 0.0009 | 3.11% |
| EAS | 0.0002 | 0.0003 | 0.62% |
| RBU | 0.0056 | 0.0032 | -14.91% |
| OTH | 0.0002 | 0.0003 | 0.62% |


**Table 2.** Trends in annual mean sulfate and BC concentrations (% per decade) in
surface air and in the column contributed by 16 anthropogenic source regions during
1980–2018 relative to the 39-year averages of total concentrations. The boldface
values are statistically significant at the 95% confidence level based on F-test.

| Region | Sulfate Conc. | Sulfate Burden | BC Conc. | BC Burden |
|--------|---------------|----------------|----------|-----------|
| Sum | **-19.83%** | **-13.18%** | **-11.93%** | **3.98%** |
| EUR | **-8.42%** | **-10.30%** | **-1.61%** | **-2.26%** |
| NAM | **-1.52%** | **-3.90%** | **0.96%** | **1.45%** |
| CAM | 0.00% | **0.05%** | 0.00% | -0.01% |
| SAM | **0.00%** | **-0.03%** | **0.00%** | **0.01%** |
| NAF | **0.02%** | **0.12%** | **0.05%** | **0.51%** |
| SAF | 0.00% | -0.02% | 0.00% | **0.18%** |
| MDE | **0.09%** | **0.85%** | **0.04%** | **0.79%** |
| SEA | 0.00% | **0.11%** | 0.00% | **0.09%** |
| CAS | **-0.72%** | **-1.01%** | **-0.04%** | -0.05% |
| SAS | **0.06%** | **3.49%** | **0.04%** | **1.97%** |
| EAS | **0.45%** | **4.24%** | **0.43%** | **5.90%** |
| RBU | **-8.54%** | **-6.64%** | **-6.12%** | **-3.74%** |
| PAN | **0.00%** | **0.00%** | 0.00% | 0.00% |
| ARC | -1.38% | -0.20% | **-5.96%** | **-1.01%** |
| ANT | 0.00% | 0.00% | **0.00%** | **0.00%** |
| OCN | 0.14% | 0.08% | **0.27%** | **0.16%** |


**Table 3.** Estimated annual mean of the response in Arctic surface temperatures (K) to
the change in TOA radiative forcing due to aerosol-radiation interactions ($RF_{ari}$) of
sulfate and BC, aerosol-cloud interactions ($RF_{aci}$) of sulfate and radiative forcing (RF)
due to BC in snow/ice (W m$^{-2}$) in each latitude band.

| Forcing location | Arctic equilibrium surface temperature response (K)* | | | |
|---|---|---|---|---|
| | Sulfate $RF_{ari}$ | Sulfate $RF_{aci}$ | BC $RF_{ari}$ | BC snow/ice |
| 60°N - 90°N | 0.035 | 0.078 | -0.005 | -0.061 |
| 28°N - 60°N | 0.059 | 0.165 | 0.029 | 0.019 |
| 28°S - 28°N | -0.001 | -0.048 | 0.031 | 0.000 |
| 90°S - 28°S | -0.005 | -0.002 | 0.002 | 0.000 |
| SUM | 0.088 | 0.193 | 0.057 | -0.041 |

*The λ are 0.31, 0.17, 0.16, 0.06 for sulfate $RF_{ari}$ and $RF_{aci}$; -0.08, 0.15, 0.31, 0.06 for
BC $RF_{ari}$; 1.06, 0.45, 0.93, 0.18 for RF due to BC in snow/ice, according to the order
given by forcing locations in the table. Sulfate $RF_{aci}$ is not archived in this study and is
roughly estimated here by scaling sulfate $RF_{ari}$ based on the ratio of sulfate $RF_{aci}$ and
$RF_{ari}$ over different latitudes from Sand et al. (2016).

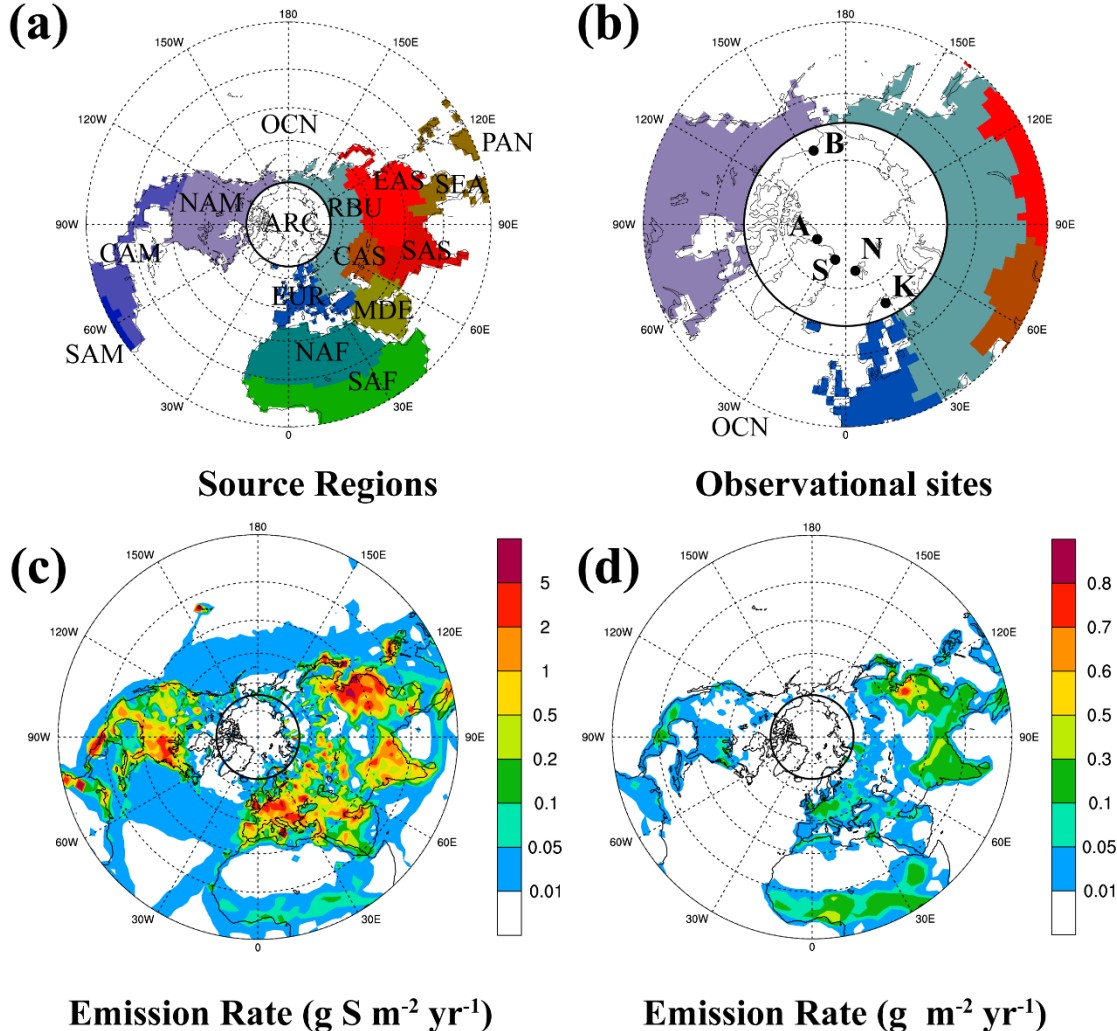


**Figure 1.** (a) Sixteen anthropogenic source regions (Europe (EUR), North America (NAM), Central America (CAM), South America (SAM), North Africa (NAF), South Africa (SAF), the Middle East (MDE), Southeast Asia (SEA), Central Asia (CAS), South Asia (SAS), East Asia (EAS), Russia-Belarus-Ukraine (RBU), Pacific-Australia-New Zealand (PAN), the Arctic (ARC), Antarctic (ANT), and Non-Arctic/Antarctic Ocean (OCN)). Dots in (b) mark observational sites at Alert ("A", 82°N, 62°W), Station Nord ("S", 81°N, 16°W), Barrow ("B", 71°N, 156°W), Ny-Alesund ("N", 78°N, 11°E) and Kevo ("K", 69°N, 27°E). Spatial distribution of annual mean (c) $SO_2$ (g S m$^{-2}$ yr$^{-1}$) and (d) BC (g C m$^{-2}$ yr$^{-1}$) emissions averaged over 1980-2018. The thick black circles mark the Arctic (66.5°N–90°N).

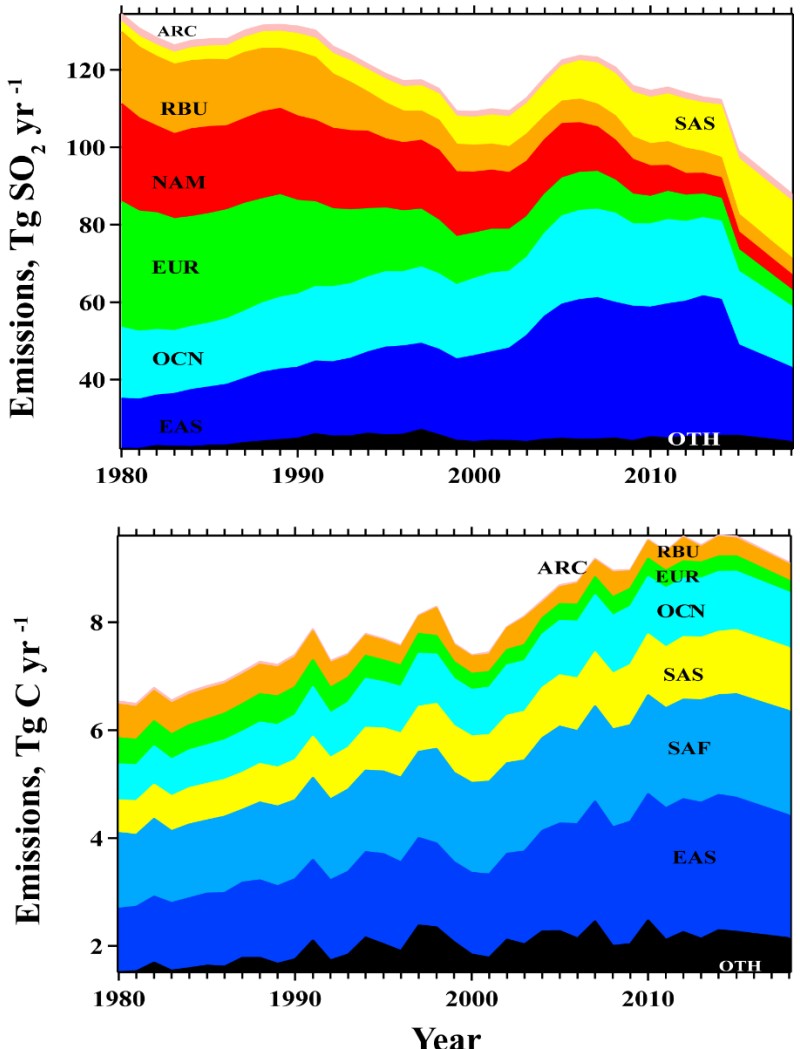

954

**Figure 2.** Time series of global total anthropogenic emissions of (top) SO₂ (Tg SO$_2$ yr$^{-1}$) and (bottom) BC (Tg C yr$^{-1}$), classified by key anthropogenic source regions. Emissions from other regions (OTH) include those of ANT, CAM, CAS, MDE, NAF, PAN, SAM, SEA, and SAF/NAM can be found in figure S1. Abbreviations for the regions can be found in Fig. 1.

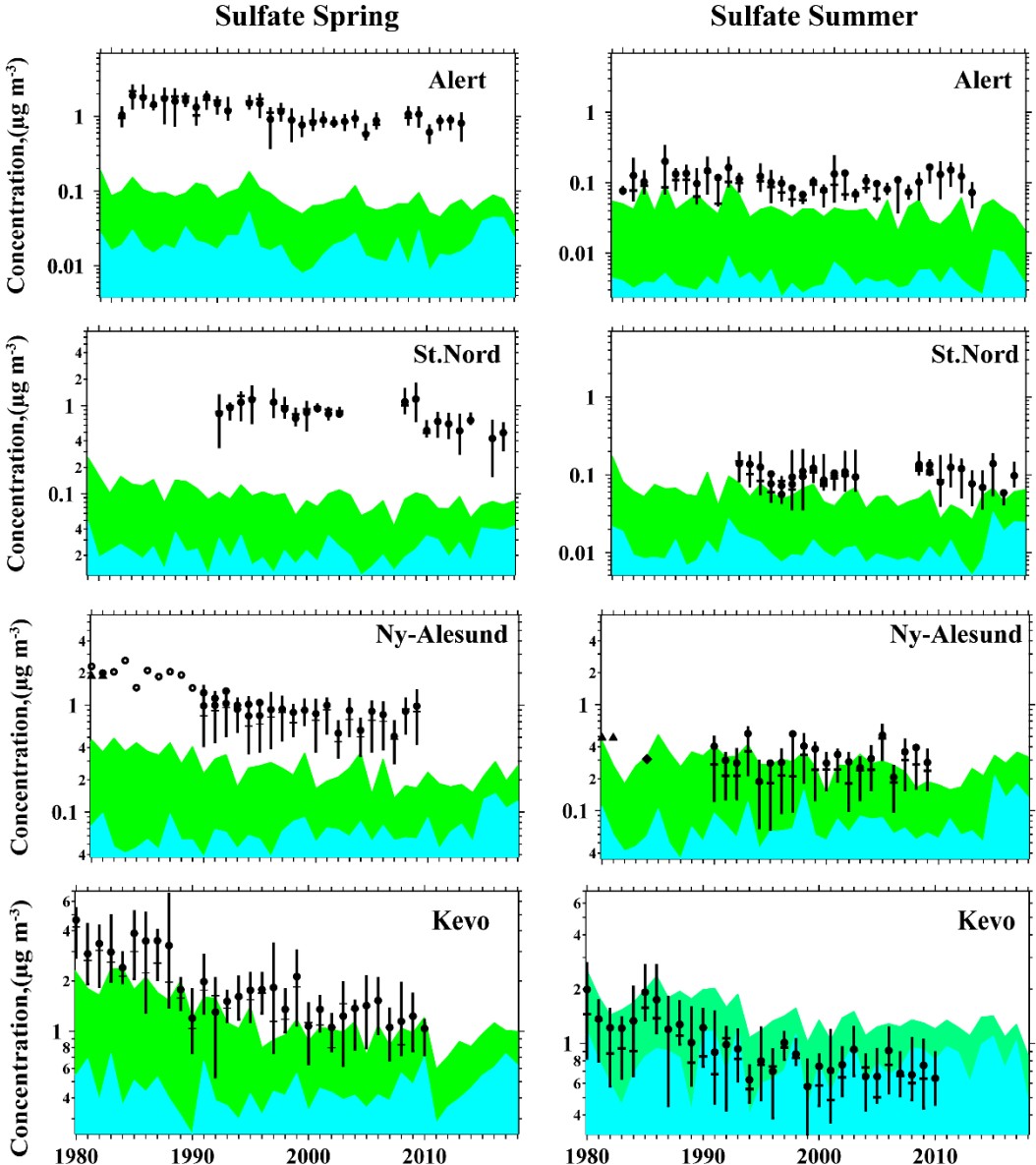

**Figure 3.** Surface concentrations of sulfate aerosols (μg m⁻³) in spring (March–May) and summer (June–August) at four locations (Alert, Station Nord, Ny-Alesund, Kevo) in the Arctic during 1980–2018. Seasonal means are denoted by solid black circles, medians as short horizontal bars, and the 25th to 75th percentile ranges as vertical bars. Stacked colors represent modeled contributions from the Arctic (blue) and non-Arctic anthropogenic source region (green). The observations denoted by solid black circles are obtained from European Monitoring and Evaluation Programme and World Data Centre for Aerosols database (http://ebas.nilu.no) and Breider et al. (2017). Black triangles at Ny-Alesund for the period 1980–1981 show mean observations from Heintzenberg and Larssen (1983). Black diamond at Ny-Alesund in summer shows median non-sea-salt sulfate concentration from Maenhaut et al. (1989). Open circles in the spring for Ny-Ålesund are March–April mean values (Sirois and Barrie, 1999). Note that the vertical coordinates use logarithmic scales.

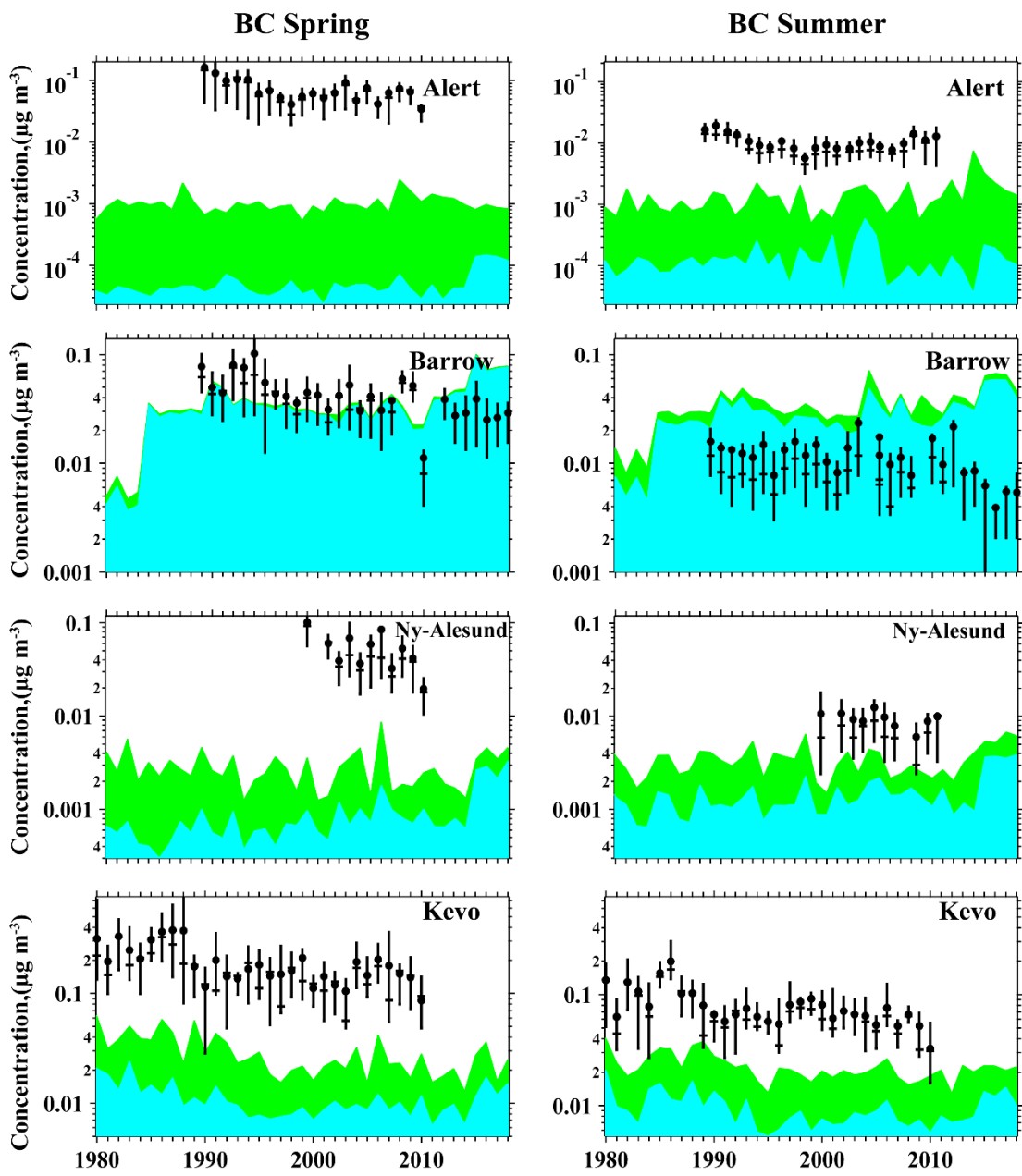


**Figure 4.** Same as Figure 3, but for surface BC (μg m⁻³) at four (Alert, Barrow, Ny-Alesund, Kevo) Arctic sites.

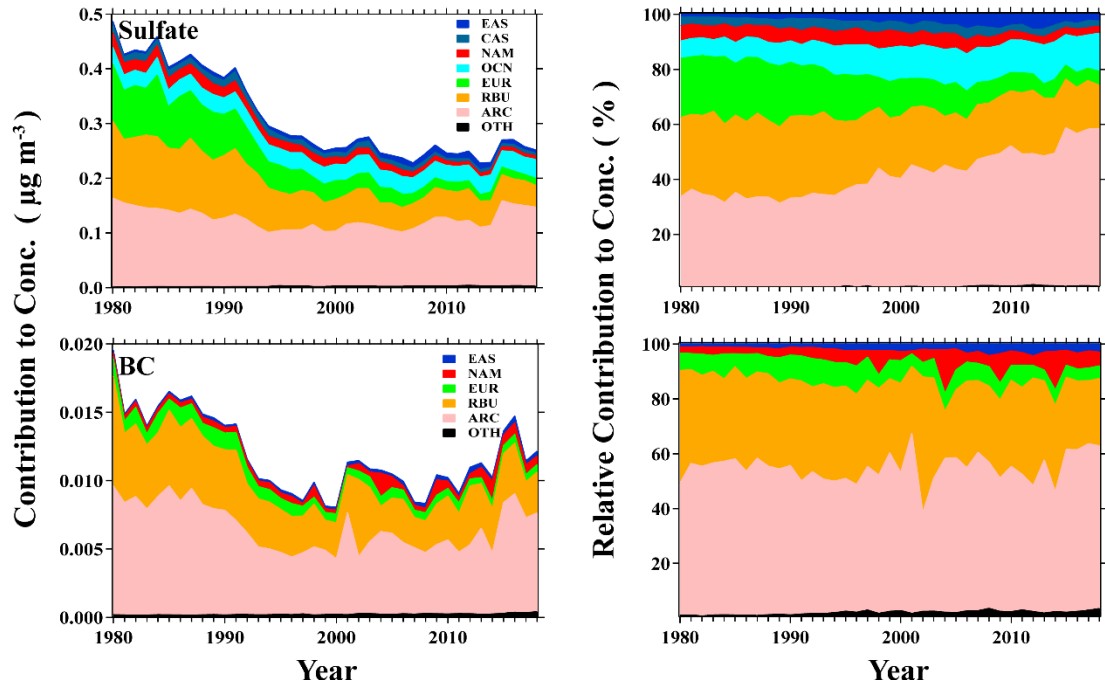


**Figure 5.** Time series (1980–2018) of absolute (left, μg m$^{-3}$) and relative (right, %)
contributions of emissions from the major source regions to the simulated annual mean
near-surface sulfate and BC concentrations averaged over the Arctic (66.5°N–90°N).
The remaining source regions with annual contributions less than 3% are combined and
shown as OTH (other regions in figure S2).

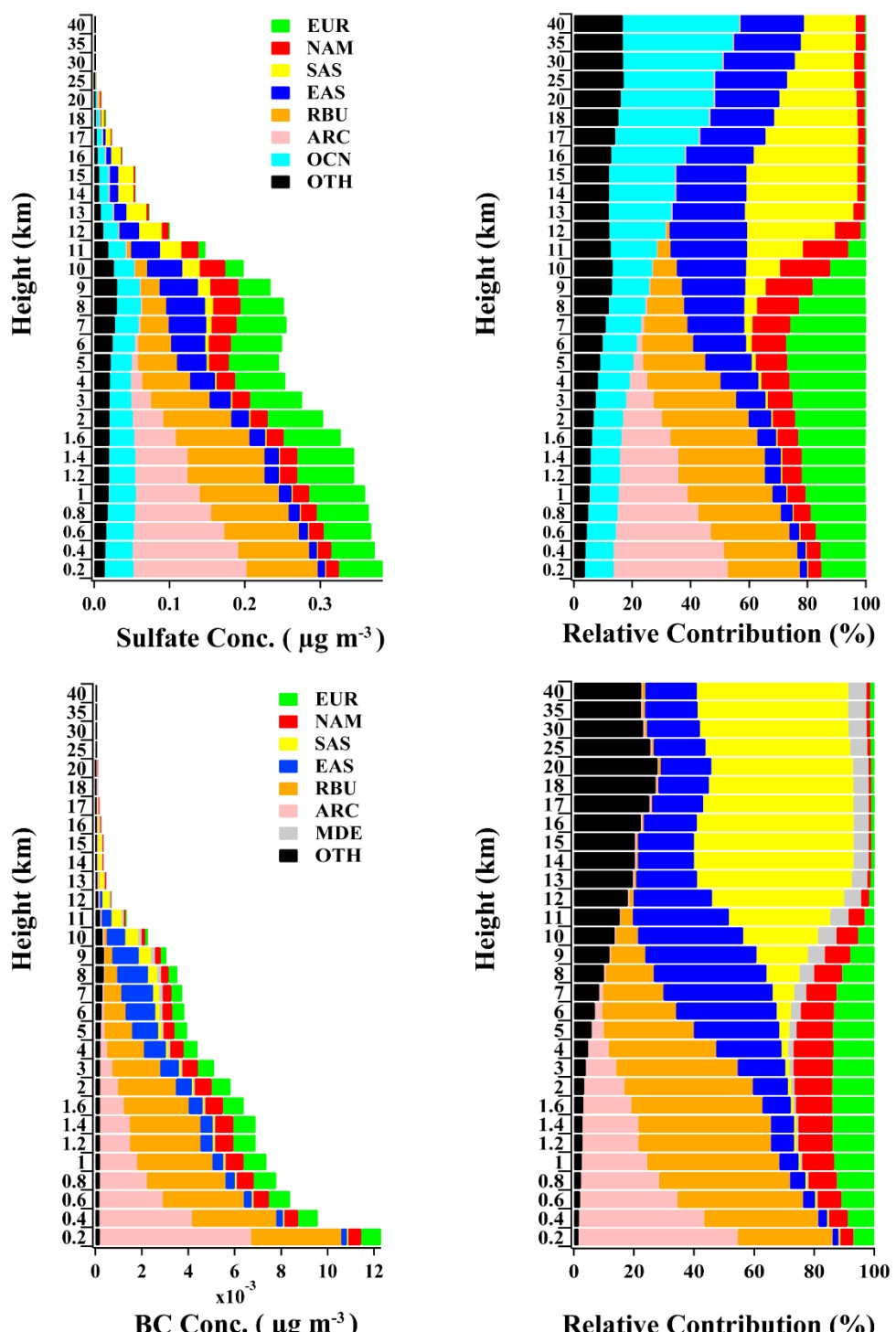

**Figure 6.** Annual mean vertical profile of sulfate (top) and BC (bottom) concentrations (μg m⁻³) over the Arctic contributed by the tagged source regions (left) and their relative contributions (right, %) during 1980–2018. Sources with annual burden contributions less than 5% are combined and shown as OTH.

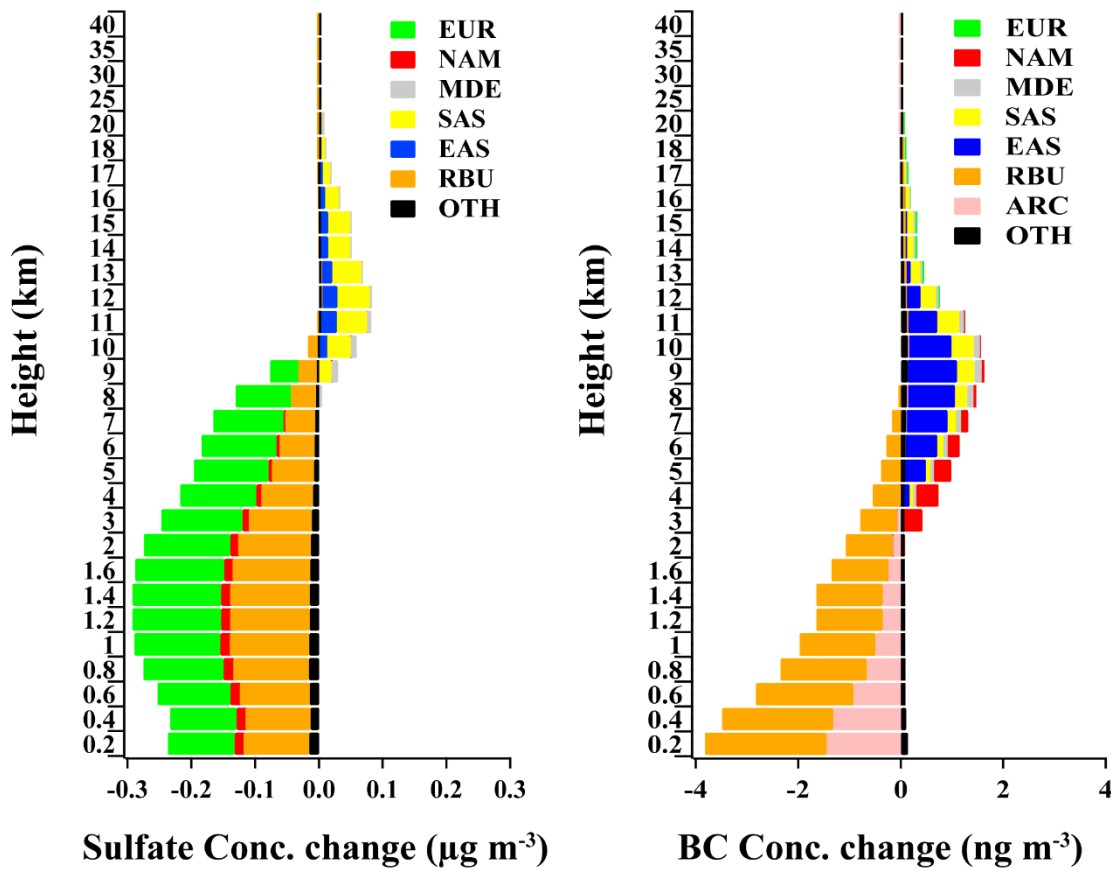

**Figure 7.** Changes in annual mean vertical profile of sulfate (μg m⁻³, left) and BC (ng m⁻³, right) concentrations over the Arctic contributed by the tagged source regions between 1980–1984 and 2014–2018.

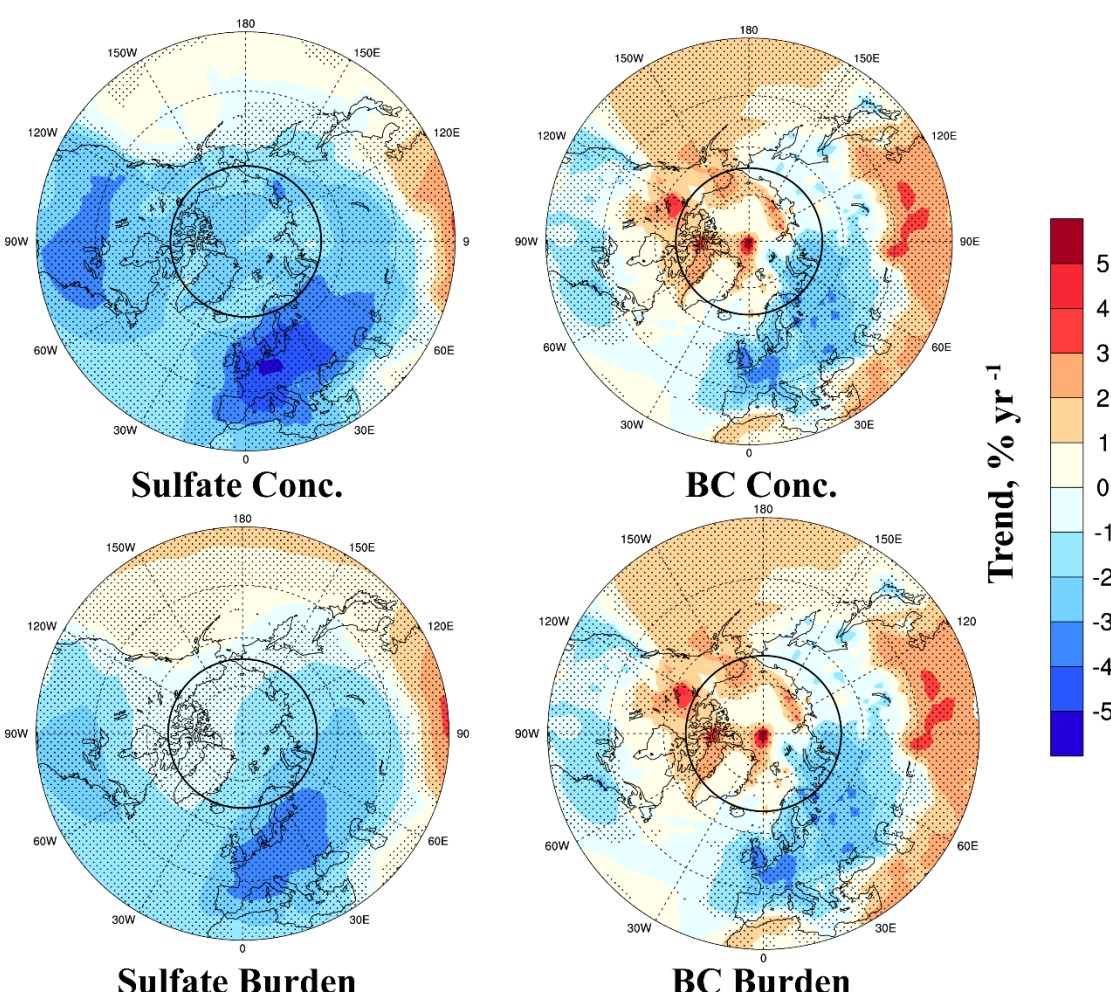


**Figure 8.** Spatial distribution of linear trends in annual mean sulfate (left) and BC (right)
concentrations (% yr⁻¹) near the surface (top) and column burden (bottom) relative to
the 39-year averages. The dotted areas indicate statistical significance with 95%
confidence based on F-test.

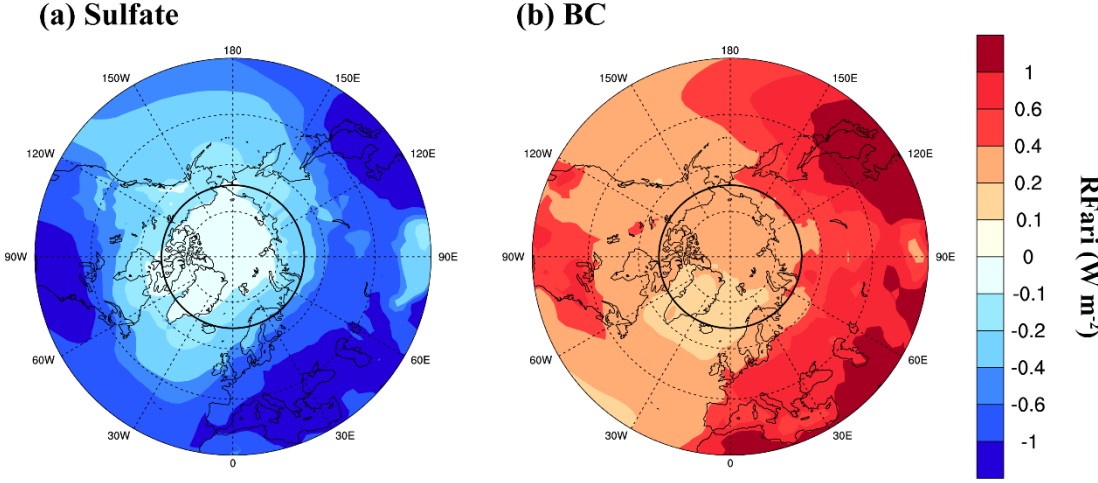

**Figure 9.** Spatial distribution of annual mean radiative forcing due to aerosol-radiation
interactions (RF$_{ari}$) of (a) sulfate and (b) BC (W m$^{-2}$) at the TOA averaged over 1980–
1001    2018.

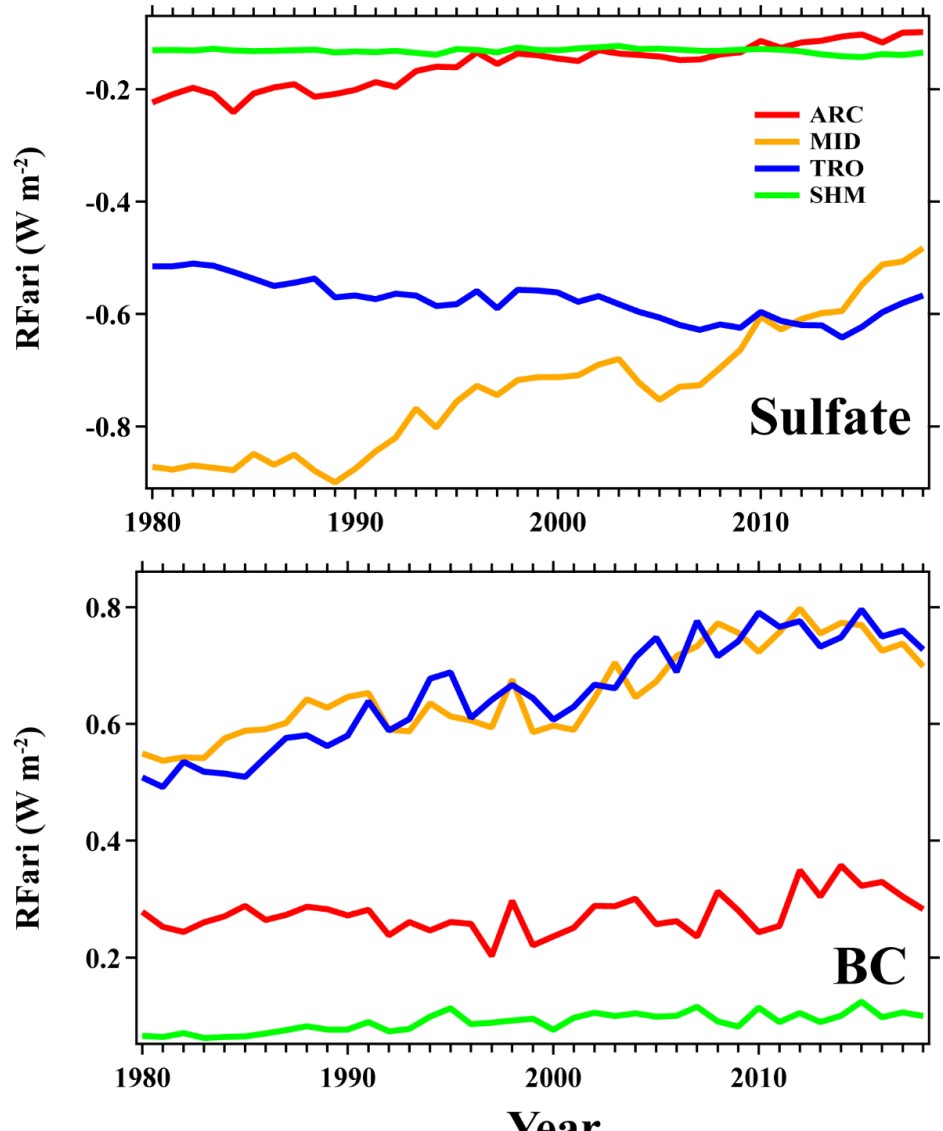

**Figure 10.** Time series (1980–2018) of annual radiative forcing due to aerosol-radiation
interactions ($RF_{ari}$, W m$^{-2}$) of sulfate and BC over the Arctic (ARC, 60°N–90°N),
Northern Hemisphere mid-latitudes (MID, 28°N–60°N), tropics (TRO, 28°S–28°N)
and Southern Hemisphere (SHM, 90°S–28°S).

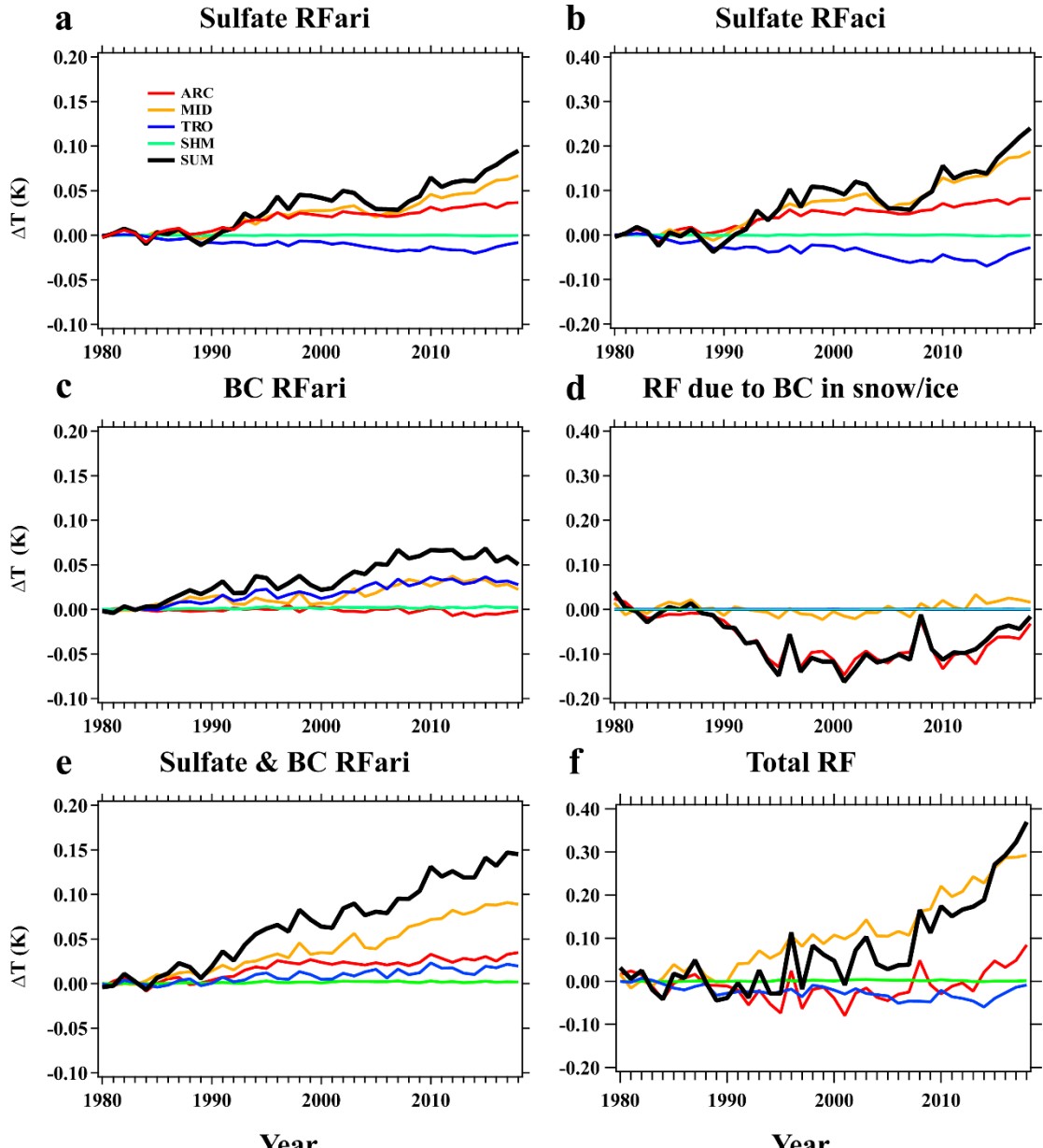


**Figure 11.** Time series (1980–2018) of the estimated response in surface temperatures (K) to the change in radiative forcing due to the aerosol-radiation interactions ($RF_{ari}$) of (a) sulfate, (c) BC, and (e) sum of sulfate and BC $RF_{ari}$, (b) radiative forcing due to aerosol-cloud interactions ($RF_{aci}$) of sulfate, (d) radiative forcing (RF) due to BC in snow/ice, (f) sum of all RF in each latitude bands and the sum of them (SUM).