# Peer review of "Source attribution of Arctic black carbon and sulfate aerosols and associated Arctic surface warming during 1980–2018"

_Atmospheric Chemistry and Physics, 2020_

## Referee Comment (RC1) · Anonymous Referee #1 · 4 Mar 2020

In this study an aerosol tagging scheme is coupled to a climate model (CAM5-EAST) to quantify the source attribution to Arctic black carbon (bc) and sulfate from 16 different regions during 1980-2018. The modelled surface concentrations are compared with measurements at four Arctic stations. Regional temperature sensitivity factors are then applied to estimate the changes in surface temperatures due to the trends in BC and SO4.

Source attribution of Arctic aerosols is a relevant scientific topic within the scope of ACP. The authors present their data in a structured way and the figures are clear. However, before consider acceptance, I recommend the authors work more on presenting their results in light of related work.

1. How does this study contribute to new knowledge in the field? What do you con-

tribute that is different (model/data set/time period)? I would highlight this in the abstract, introduction and conclusion.

2. Your conclusions are not new (but it is still very important to test what others have done!), but I would then add, 'as also shown in . . .etc etc. For instance, it have been shown in other studies that the declining emissions in Europe and the collapse of the Soviet Union are the main reasons why we see declining trends in the Arctic and that emissions from Asia contribute to higher level aerosols in the high-Arctic.

3. I would also compare your numbers with other studies. Do they differ from other studies or do they support other findings? If different; try to explain why.

4. The authors use sensitivity factors to estimate the temperature response to the declining trends. This method needs to be explained in Methods along with uncertainties.

5. Can you please add a description in Methods on how BC and sulfate are treated in the model? Aging, mixing etc.

Specific comments by line number:

Title: You are only looking at BC and SO4, so I would change 'aerosols' to reflect that + specify surface warming, and not just warming.

L23: Wouldn't a decrease in BC, at least hypothetically, lead to a cooling?

L32: You need to specify that you have calculated the surface temperature response using sensitivity factors (and not by running a climate model).

L42: What other regions do you refer to here? Most aerosols are emitted NH mid lats?

L140: What kind of aerosol-cloud interaction are included in the model?

L197: Where in the Arctic are those emissions mostly from? I would assume northern Russia?

L226: Could you be more specific on where the Kevo site is located besides close to

western Eurasia?

L217: Can you split these two sentences; one for bc and one for sulfate so it is easier to follow?

L257: Could you remind us which regions those are?

L280: this is the first time you report concentrations in ug/m3 decrease and not % decrease. Can you add the total concentration number as well, so we can relate the number?

L313: 'during'? How is this calculated? First and last 5 years?

L329: What is a moderate value?

L332: this is the first time you mention the tropical region?

L348: I would decrease the number of significant figures for these temperature response numbers, as the uncertainties are much higher.

L394: Can you list these references you refer to here?

L400: 'to some extent' seem vague.

L408: Increase compared to what?

Figure 1: it is hard to see the letters/dots representing the observation sites. Could another plot be made in this figure, zooming in on the Arctic (90-60N) and only showing the stations for example?

---

## Referee Comment (RC2) · Anonymous Referee #2 · 10 Mar 2020

This study uses source apportionment method to study the changes Arctic BC and Sulfate concentration, and the contributions from worldwide 16 other regions. They also performed sensitivity analysis to discuss the contribution of Arctic warming from the different source regions.

In general, I think the paper has an interesting theme. However, the method is not well presented, and the discussion is not well structured neither. The paper heavily focusses on the model results, and was not strong to make adequate discussions on why the simulated results happen.

Main comment:

I suggest the authors reorganize the abstract from L32-43: think about the order of

discussing the sulfate/BC radiative forcing changes, local vs long-range transport, temperature changes from aerosol-direct and indirect effects.

It has been known that there are very large discrepancies for the emissions in China from MEIC emission inventory and CMIP6 (Paulot et al., 2018). Comment how this discrepancy could affect the main results.

Reference: Paulot, F., Paynter, D., Ginoux, P., Naik, V., and Horowitz, L. W.: Changes in the aerosol direct radiative forcing from 2001 to 2015: observational constraints and regional mechanisms, Atmos. Chem. Phys., 18, 13265–13281, https://doi.org/10.5194/acp-18-13265-2018, 2018.

Beginning from section 3, when the authors discuss the trends analysis, I did not find anywhere how the authors performed the trend analysis, as well as the significance test. Those are very basic concepts when we discuss trend analysis. A few example: line 245-line 248; line 251-252, and Table 2, Fig. 8.

Line 269-270: when the authors discuss the "largest contribution of East and South Asia", does the authors mean East and South Asia contributes most at this altitude compared with other regions, or this altitude is where East and South Asia contributes most for their contributions at different altitudes? As a matter of fact, I think these several paragraphs are terribly written (line 264-290). Keep in mind that, when you talk about the contribution, you are comparing between different source regions as well as the altitudes. I highly suggest the authors reorganize these several paragraphs.

Editorial comments: Line 35: explain what "61%" is compared to.

Line 38: the snow/ice albedo effect from BC refers to local or other source regions?

Line 98: add from which year for the 2-3% changes.

Line 122: change "observational" to "observation"

Line 153: EAST was already defined.

[Figure]

Line 181-182: Technically, neither Fig 1 nor Fig 2 showed the emission changes from "1980-2010" "from the 16 source regions".

Line 209-210: Please put reference or show precipitation/wet deposition plots to confirm the theory.

Line 212-213: I thought local BC reduction by 38% in Artic are pretty high. So are you sure the BC concentration changes are dominated by the emission changes from other source region? Meanwhile, I got different conclusion from Fig. 5 as the ARC clearly dominated the total BC changes.

Line 256: how did the "+/- 1-3%" come from? It looks like uncertainty range to me.

Line 262-263: the authors conclude to reduce local sources in the Arctic to control the sulfate and BC. Can the authors give some specific suggestions on the sectors which the local source should be reduced?

Line 272-273: If I am reading the plots right, I think Europe also has largest contribution for both sulfate and BC below 2km, compared with Arctic? That being said, I still can not figure out what the authors refer to when they say "largest contribution".

Line 287-288: Again not clear how to comprehensive the "increasing trend" contributed by East Asia and South Asia. Also, the authors have a theory why East Asia and South Asia are larger high altitudes, any references or evidences? Please explain.

Line 317-322 Here is redundant to discuss the radiative forcing changes in other NH regions since this is not the focus of this paper. Remember the paper's interest is on the transport of other source region on the reception region (Arctic).

Line 329-330: The authors previously showed that the sulfate concentration changes over Arctic are dominated by other source regions than local. So why the authors conclude the local sulfate change for the radiative forcing increases?

Line 330-333: again these are not relevant to this study. I think it is Ok if the authors

want to compare the radiative forcing changes in Artic with other regions for the past 4 decades, but not necessary to distract the main point of the paper.

Line 356-358: please explain why the BC changes over mid-latitude and tropics have positive climate effect and expand to Arctic?

Figures:

In Fig. 2 title, add the references that abbreviations for the regions could be found in Fig. 1

Fig. 3 I saw crosses, triangles, rectangles and dotted circles which are not explained in the legend. In the stacked contour plots, I think the authors refer light green for the Arctic? The Y axis for plots St. Nord Ny-Alesund and Kevo seems not right to me.

In Fig 5 and figures below, the authors only show a list of the source regions, not all of them. I suspect that's because other regions' contribution to BC and sulfate in Arctic are very negligible? If so, how much is it? Is it magnitude level smaller than the CAS to sulfate, and EAS to BC? Also, how did the authors make the relative contribution equal to 100% if not all the regions included? I would also suggest the authors to reorganize the plot, so maybe the contour plots will be seen as smaller to largest, or vice-visa.

Fig 9: this study's focus is on Arctic. This fig is not easy to distinguish the spatial patterns of temperature changes in Arctic.

---

## Referee Comment (RC3) · Anonymous Referee #3 · 20 Mar 2020

Review of "Source attribution of Arctic aerosols and associated Arctic warming trend during 1980-2018" by Ren et al.

This paper presents a modelling study of the impacts of changing SO4 and BC on the Arctic atmospheric composition, radiative forcing, and temperature. Modelled and measured SO4 and BC are presented in the Arctic from 1980-2018 at a handful of surface measurement sites. A tagged version of CAM5 is used to quantify the source contributions from different continental geographic regions to the Arctic BC and SO4 concentrations both at the surface and in the vertical column. The paper present interesting results that are important for understanding the rapidly warming Arctic. The authors conclude that about 20% of Arctic warming can be attributed to the combination of BC and SO4. I suggest only the following minor revisions below before publishing:
lines 130-131: is there a primary reference for CAM5 and CESM that you can reference here?

lines 143-144: what is the source for the specified sea surface temperatures, sea ice concentrations, etc?

lines 209-210: was the modelled precipitation compared to measured precipitation? Was wet deposition of model validated against measurements?

Fig 5/line 241: it needs to be clarified that Fig 5 is the model average in the Arctic ($>66.5°$N).

line 252: was that rise in BC seen in the observations? e.g., consistent with BC seen at Alert?

line 263: "in the Arctic" ... and Russia?

line 316: is the effect of BC deposition on snow/reduction of albedo included in this? I think not because that effect is discussed later, but could clarify here that this value is just for atmospheric BC effect.

Section 5/line 400: Can you add some discussion as to how the model bias affects your conclusions? E.g. would your estimates of SO4 and BC temperature impacts be greater or lesser if the model were corrected to accurately reflect the measurements?

Data availability: please add where the Arctic BC & SO4 measurements can be found in this section (e.g., EBAS database link).

Figs 1-2, and 5-7: please make sure the regional colours are consistent in all of these plots. e.g., colour X for RBU, colour Y for EUR, etc, in all 5 figures the same.

Fig 3 (4): Clarify in the caption that the black is from measurements, and the blue and green are modelled. E.g., "*Measured* seasonal means are denoted by...". "Stacked contours represent the *modelled* Arctic..."

Fig 3: why is Barrow not shown? Fig 4: why is St Nord not shown? Fig 5: specify that this is the Arctic ($>66.5°$N) average. As mentioned above, use the same regional colour scheme here as in Fig 1(a) & Fig 2. Fig 6 & 7: match the regional colours to Fig 5.

---

## Author Comment (AC1) · 26 May 2020

**Manuscript # acp-2020-3**

**Responses to Referee #1**

Source attribution of Arctic aerosols is a relevant scientific topic within the scope of ACP. The authors present their data in a structured way and the figures are clear. However, before consider acceptance, I recommend the authors work more on presenting their results in light of related work.

We thank the reviewer for all the insightful comments. Below, please see our point-by-point response (in blue) to the specific comments and suggestions and the changes that have been made to the manuscript, in an effort to take into account all the comments raised here.

1.  How does this study contribute to new knowledge in the field? What do you contribute that is different (model/data set/time period)? I would highlight this in the abstract, introduction and conclusion.

Response:
    Thanks for the suggestion. Many studies have examined possible mechanisms that can explain the recent Arctic warming, but the quantitative importance of these mechanisms is still on debate. Among these mechanisms, some are related to roles of aerosols in changing the Arctic temperature. Shindell and Faluvegi (2009) found that aerosols may have warmed the Arctic surface due to emission reductions during 1976-2010. Breider et al. (2017) estimated that emission reductions in anthropogenic aerosols during 1980–2010 had contributed to a net warming at the Arctic surface by +0.27 ± 0.04 K using the GEOS-Chem model, which is consistent with our results. However, they did not take into consideration of the radiative forcing from aerosol-cloud interactions and deposition of BC to snow and ice surfaces. Navarro et al. (2016) presented simulations with an Earth system model and showed that the reduction in European $SO_2$ emission over 1980–2005 has caused an Arctic warming by 0.5 K on annual average as a result of the enhanced poleward heat transport, which is larger than our estimates likely due to different emissions and models used here and in Navarro et al. (2016).
    Different from the emission perturbation method that was often used in previous studies, in this study, a global aerosol-climate model equipped with an Explicit Aerosol Source Tagging (CAM5-EAST) is applied to quantify the source apportionment of aerosols in the Arctic from sixteen source regions and the role of aerosol variations in affecting changes in the Arctic surface temperature from 1980 to 2018. All aerosol radiative impacts are considered including aerosol-radiation and aerosol-cloud interactions, as well as black carbon deposition on snow and ice-covered surfaces. We quantitatively showed that the combined

total effects of sulfate and BC produced an Arctic surface warming of +0.297 K, explaining approximately 20% of the observed Arctic warming. We have now highlighted these in the various components of the manuscript.

2. Your conclusions are not new (but it is still very important to test what others have done!), but I would then add, 'as also shown in…etc etc. For instance, it have been shown in other studies that the declining emissions in Europe and the collapse of the Soviet Union are the main reasons why we see declining trends in the Arctic and that emissions from Asia contribute to higher level aerosols in the high-Arctic.

Response:
Thanks for the suggestion. We have now included such context for our conclusions as follows: "Previous studies also pointed out that, in April, BC showed a high concentration in the mid-troposphere of the Arctic, mainly due to the effect of Asian anthropogenic aerosols that are transported to the Arctic through warm conveyor belt (Wang et al., 2011). Evidence from aircraft and ground-based measurements showed that eastern and southern Asia source regions contributed the most to the BC concentration in the Arctic mid-troposphere, while northern Asia dominated the contribution to the Arctic surface BC (Abbatt et al., 2019)." And "Similar to our findings, Breider et al. (2017) found that the simulated decrease in aerosol optical depth in the Arctic from 1980 to 2010 was driven by a strong decrease in aerosol loading at lower altitudes due to the emission changes in West Eurasia, Russia and North America and an increase in aerosols at higher altitudes resulting from the changes in emissions in regions such as South Asia and East Asia."

3. I would also compare your numbers with other studies. Do they differ from other studies or do they support other findings? If different; try to explain why.

Response:
Thanks for the suggestion. The atmospheric BC can absorb solar radiation in the atmosphere and leads to a positive $RF_{ari}$ of 0.1~0.4 $Wm^{-2}$ in the Arctic, which is similar to the values of 0.1~0.6 $Wm^{-2}$ estimated in previous studies (Koch and Hansen, 2005; Flanner et al., 2009; AMAP, 2011; Bond et al., 2011; Samset et al., 2014; Wang et al., 2014).
Shindell et al. (2008) studied the sensitivity of simulated Arctic aerosol concentrations to emissions perturbations in 2001 and found that European emissions contributed to Arctic sulfate concentrations near the surface and at 500 hPa by 73% and 51%, respectively. East Asia has the largest contribution at 250 hPa, reaching 36%, which is consistent with our results. Based on simulations of a chemical transport model, Fisher et al. (2011) concluded that West Asia emissions dominated wintertime Arctic sulfate concentration, with

contributions between 30% and 45%.

Using the GEOS-Chem model, Breider et al. (2017) estimated that emission reductions in anthropogenic aerosols during 1980–2010 had contributed to a net warming at the Arctic surface by +0.27 ± 0.04 K, which is consistent with our results. However, they did not take into consideration radiative forcing from aerosol-cloud interactions and deposition of BC to snow or ice surfaces. Navarro et al. (2016) presented simulations with an Earth system model and showed that the reduction in Europe $SO_2$ emission over 1980–2005 has caused the Arctic warms by 0.5 K on annual average as a result of the enhanced poleward heat transport, which is larger than our estimates likely due to different emissions and models used here and in Navarro et al. (2016).

We have included these comparisons in the manuscript.

4. The authors use sensitivity factors to estimate the temperature response to the declining trends. This method needs to be explained in Methods along with uncertainties.

Response:

Thanks for the suggestion. The Arctic equilibrium temperature response is estimated using Arctic climate sensitivity factors ($\lambda$, K $W^{-1}m^2$), defined as the change in Arctic surface temperature per unit RF for different latitudinal bands from Sand et al. (2016) and Shindell and Faluvegi (2009). The change in equilibrium temperature response is defined as $\Delta T = \sum_{j=LAT} \lambda_j * \Delta RF_j$ . $\Delta$ represents the difference of the annual mean of a variable for a specific year compared to the average during 1980–1984 in this study. RF is radiative forcing due to aerosol-radiation or aerosol-cloud interactions associated with sulfate or black carbon. LAT represents latitudinal bands over the Arctic (60°N–90°N), Northern Hemisphere mid-latitudes (28°N–60°N), tropics (28°S–28°N) and Southern Hemisphere (90°S–28°S). Many studies used these climate sensitivity factors to estimate the Arctic temperature responses using RF calculated from different models (e.g., Sand et al., 2016). However, we note that, since the $\lambda$ values were calculated with a different climate model (NASA-GISS), the estimated Arctic equilibrium temperature response based on these factors could be biased.

5. Can you please add a description in Methods on how BC and sulfate are treated in the model? Aging, mixing etc.

Response:

Thanks for the suggestion. We have now added a description as follows. Mass and number concentrations of sulfate particles are predicted for the three lognormal modes (i.e., Aitken, accumulation, and coarse modes) of the three-mode modal aerosol module (Liu et al., 2012) in CAM5. Aerosols are internally

mixed in the same aerosol mode and then externally mixed between modes. Within each mode, sulfate is internally mixed with primary/secondary organic matter, BC, mineral dust, and/or sea salt. BC is mixed with other aerosol species (e.g., sulfate, POA, SOA, sea salt, and dust) in the accumulation mode immediately after being emitted into the atmosphere without considering explicit aging processes.

Specific comments by line number:

Title: You are only looking at BC and SO4, so I would change 'aerosols' to reflect that + specify surface warming, and not just warming.

Thanks for the suggestion. We have now modified the title to "Source attribution of Arctic black carbon and sulfate aerosols and associated Arctic surface warming during 1980–2018."

L23: Wouldn't a decrease in BC, at least hypothetically, lead to a cooling?

That's likely true for the Arctic local BC alone. To avoid confusion, this sentence has been revised as follows: "Observations show that the concentrations of Arctic sulfate and black carbon (BC) aerosols have declined since the early 1980s. Previous studies have reported that reducing sulfate aerosols potentially contributed to the recent rapid Arctic warming."

L32: You need to specify that you have calculated the surface temperature response using sensitivity factors (and not by running a climate model).

We have now added "By using climate sensitivity factors, …".

L42: What other regions do you refer to here? Most aerosols are emitted NH mid lats?

The other regions refer to latitudinal bands: Arctic (ARC, 60°N–90°N), tropics (TRO, 28°S–28°N) and Southern Hemisphere (SHM, 90°S–28°S). Aerosols over any region can influence Arctic surface temperature through changing radiative fluxes or poleward heat transport based on the climate sensitivity factors. The mid-latitude region of the Northern Hemisphere is close to the Arctic and changes in aerosols over this region affect Arctic temperature through enhancing poleward heat transport. This warming effect is stronger than impacts of aerosols over other latitudinal bands.

L140: What kind of aerosol-cloud interaction are included in the model?

Aerosols interact with stratiform clouds through two-moment microphysics, in which the nucleation of stratiform cloud droplets is based on the scheme of Abdul-Razzak and Ghan et al. (2000). Although aerosols have no microphysical impact on convective clouds, the ambient temperature and convection can be affected by BC-induced atmospheric heating. We have added this description in the Methodology section.

L197: Where in the Arctic are those emissions mostly from? I would assume northern Russia?

Time series (1980–2018) of absolute and relative contributions of emissions from major source regions to the simulated annual mean near-surface sulfate and BC concentrations averaged over the Arctic is shown in Figure 5. It's correct that source regions near the Arctic (e.g., Europe and Russia) are the main contributors to the near-surface concentrations of Arctic sulfate and BC.

L216: Could you be more specific on where the Kevo site is located besides close to western Eurasia?

We have now revised the sentence to reflect this: "The Kevo site (69°N, 27°E), which is close to Western Eurasia, is the only site that has both sulfate and BC data for more than 30 years."

L217: Can you split these two sentences; one for bc and one for sulfate so it is easier to follow?

Following the suggestion, we have split the sentences for BC and sulfate: "At this site, the simulated sulfate in spring and summer decreased at a rate of -3.18% and -1.92% per year, respectively, which are similar to -4.37% and -3.26% per year from observations. The decreasing rates of BC in spring and summer were -2.89% and -1.74%, respectively, that are also consistent with the observed values of - 3.01% and - 2.82%."

L257: Could you remind us which regions those are?

The remaining source regions are Central America (CAM), South America (SAM), North Africa (NAF), South Africa (SAF), the Middle East (MDE), Southeast Asia (SEA), Central Asia (CAS), South Asia (SAS), Pacific-Australia-New Zealand (PAN), Antarctic (ANT), and Non-Arctic/Antarctic Ocean (OCN). We have now included such information in the revised text.

L280: this is the first time you report concentrations in ug/m3 decrease and not % decrease. Can you add the total concentration number as well, so we can relate the number?

Following the suggestion, we have now revised the text as follows:

"Relative to the average of 0.447 µg/m$^3$ during 1980–1984, the simulated annual sulfate concentration over the Arctic has a decrease of 42.8% (0.191 µg/m$^3$) in 2014–2018 (Table 1). Sulfate concentration shows a considerable decreasing trend from 1980 to 2000, which then slows down after 2000. The decrease in sulfate during this time period primarily results from the reduction in emissions from Europe and Russia, which contributes to 18.6% (0.083 µg/m$^3$) and 18.8% (0.084 µg/m$^3$) of the decline of the Arctic sulfate concentrations, respectively. The change in emissions from Central Asia and North America, respectively, explains 1.6% (0.007 µg/m$^3$) and 3.4% (0.015 µg/m$^3$) of the reduced concentration."

"Simulated Arctic BC concentration also shows a considerable decline before 2000, but a slight rise after 2000. Overall, the average concentration of BC in the Arctic had a decrease of 22.98% (3.7 ng/m$^3$ relative to the 1980–1984 average of 16.1 ng/m$^3$) in 2014–2018, mainly due to the reductions in emissions originating from the Arctic and Russia, which lead to 9.32% (1.5 ng/m$^3$) and 14.91% (2.4 ng/m$^3$) of the decrease (Table 1)."

L313: 'during'? How is this calculated? First and last 5 years?
    We have revised it to "averaged over 1980–2018".

L329: What is a moderate value?
    We have revised the text as follows: "Within the Arctic (60°N–90°N), the magnitude of sulfate RF$_{ari}$ decreases from -0.21 Wm$^{-2}$ in 1980–1984 to -0.10 Wm$^{-2}$ in 2014–2018, indicating a warming effect in the Arctic from the local sulfate change."

L332: this is the first time you mention the tropical region?
    Yes. To estimate the relative roles of regional aerosol trends in affecting the Arctic warming, we looked into the temporal variation of annual mean radiative forcing of sulfate and BC in different latitudinal bands during 1980–2018. The four latitudinal bands considered in this study are Arctic (60°N–90°N), Northern Hemisphere mid-latitudes (28°N–60°N), tropics (28°S–28°N) and Southern Hemisphere (90°S–28°S).

L348: I would decrease the number of significant figures for these temperature response numbers, as the uncertainties are much higher.
    We agree with the reviewer that the uncertainties associated with these numbers are likely high, but the number of digits after the decimal point is kept same for all the numbers here for consistency.

L394: Can you list these references you refer to here?
    Many studies have examined possible mechanisms that can explain the recent Arctic warming, but the quantitative importance of these mechanisms is still on debate (e.g., Breider et al., 2017; Navarro et al. 2016).

L400: 'to some extent' seem vague.
    We have revised the text as follows: "Considering that the model underestimates the magnitude of sulfate and BC concentrations, the estimated impact on Arctic temperature from sulfate and BC could be even larger if the model were able to accurately reproduces the measurements in the Arctic."

L408: Increase compared to what?
    We have revised it to "Compared to the annual mean concentrations during 1980–1984".

Figure 1: it is hard to see the letters/dots representing the observation sites. Could another plot be made in this figure, zooming in on the Arctic (90-60N) and only showing the stations for example?

Following the suggestion, we have now revised the Figure 1 to zoom in to the Arctic for a better display of the observational sites. Please see below.

[revised manuscript text omitted]

---

## Author Comment (AC2) · 26 May 2020

| 1        | Manuscript # acp-2020-3                                                                           |
|----------|---------------------------------------------------------------------------------------------------|
| 2        | Posponsos to Poforoo #2                                                                           |
| 3        | Responses to Referee #2                                                                           |
| 4
5   | This study uses source apportionment method to study the changes Arctic BC                        |
| 6        | and Sulfate concentration, and the contributions from worldwide 16 other                          |
| 7        | regions. They also performed sensitivity analysis to discuss the contribution of                  |
| 8        | Arctic warming from the different source regions.                                                 |
| 9        | In general, I think the paper has an interesting theme. However, the method is                    |
| 10       | not well presented, and the discussion is not well structured neither. The paper                  |
| 11       | heavily focusses on the model results, and was not strong to make adequate                        |
| 12       | discussions on why the simulated results happen.                                                  |
| 13       | We thank the reviewer for all the insightful comments. Below, please see our                      |
| 14       | point-by-point response (in blue) to the specific comments and suggestions                        |
| 15       | and the changes that have been made to the manuscript, in an effort to take                       |
| 10       | into account all the comments raised here.                                                        |
| 10       | Main comment:                                                                                     |
| 10       | I suggest the authors reorganize the abstract from 1 32-43: think about the order                 |
| 20       | of discussing the sulfate/BC radiative forcing changes, local vs long-range                       |
| 21       | transport, temperature changes from aerosol-direct and indirect effects.                          |
| 22       | Response:                                                                                         |
| 23       | Following the suggestion, we have now revised this part of the abstract as                        |
| 24       | follows: "Within the Arctic, sulfate reductions caused a TOA warming of 0.11                      |
| 25       | and 0.25 W m -2 , respectively, through aerosol-radiation and aerosol-cloud            |
| 26       | interactions. While the changes in Arctic atmospheric BC has little impact on                     |
| 27       | local radiative forcing, the decrease of BC in snow/ice led to a net cooling of                   |
| 28       | 0.05 W m -2 . By applying climate sensitivity factors for different latitudinal bands, |
| 29       | global changes in sulfate and BC during 2014–2018 (with respect to 1980–1984)                     |
| 30       | exerted a +0.088 K and 0.057 K Arctic surface warming, respectively, through                      |
| 31       | aerosol-radiation interactions. Through aerosol-cloud interactions, the suifate                   |
| 3Z
22 | The weakened BC effect on snow/ice albede led to an Arctic surface cooling of                     |
| 33
34 | -0.041 K. The changes in atmospheric sulfate and BC outside the Arctic totally                    |
| 34       | produced an Arctic warming of $\pm 0.25$ K the majority of which is due to the mid-               |
| 36       | latitude changes in radiative forcing. Our results suggest that changes in                        |
| 37       | aerosols over the mid-latitudes of the Northern Hemisphere have a larger                          |
| 38       | impact on Arctic temperature than other regions through enhanced poleward                         |
| 39       | heat transport. The combined total effects of sulfate and BC produced an Arctic                   |
| 40       | surface warming of +0.297 K, explaining approximately 20% of the observed                         |
| 41       | Arctic warming since the early 1980s."                                                            |
| 42       |                                                                                                   |

It has been known that there are very large discrepancies for the emissions in
China from MEIC emission inventory and CMIP6 (Paulot et al., 2018).
Comment how this discrepancy could affect the main results.

46 Reference: Paulot, F., Paynter, D., Ginoux, P., Naik, V., and Horowitz, L. W.:

Changes in the aerosol direct radiative forcing from 2001 to 2015: observational
constraints and regional mechanisms, Atmos. Chem. Phys., 18, 13265–13281,

49 https://doi.org/10.5194/acp-18-13265-2018, 2018.

50 Response:

Thanks for bringing up this issue. In our simulations of 1980-2018, we used 51 both the CMIP6 historical emissions for 1980-2014 and emission scenario 52 (SSP2-4.5) interpolated 2015-2018. Over China, the decline of aerosols 53 emissions since 2011 is not well represented in the CMIP6 historical 54 55 anthropogenic emissions, compared to the MEIC emission inventory (Paulot et al., 2018). Emissions of SO2 and BC from China in SSP2-4.5 show declines 56 since 2014, which is consistent with MEIC emissions. However, the decrease 57 of CMIP6 SO2 and BC emissions over China by 39% and 0.5%, respectively, 58 59 in year 2017 compared to 2010 is less than the corresponding magnitude, 62% 60 and 27%, in MEIC emission inventory. We have now included this point in the discussion section as follows: "Previous studies have reported large 61 discrepancies of aerosol and precursors emissions in China between MEIC 62 (Multi-resolution Emission Inventory for China) and CMIP6 emission inventories 63 (e.g., Paulot et al., 2018). The CMIP6 emissions dataset shows similar 64 decreasing trends in anthropogenic SO2 and BC emissions over China since 65 2011 as in the MEIC inventory (Fig. S3). However, the decrease of CMIP6 66 anthropogenic SO2 and BC emissions by 39% and 0.5%, respectively, in 2017 67 compared to 2010 is less than the corresponding magnitude of 62% and 27% 68 in MEIC (Zheng et al., 2018). It indicates that the increase in aerosol 69 contribution from East Asia during the recent decade and its impact on Arctic 70 71 surface temperature could be overestimated in this study."

72

Figure S3. Annual anthropogenic emissions of SO2 and BC in China from
 CMIP6 (solid lines) and MEIC (dotted lines).

76

Beginning from section 3, when the authors discuss the trends analysis, I did not find anywhere how the authors performed the trend analysis, as well as the significance test. Those are very basic concepts when we discuss trend analysis. A few example: line 245-line 248; line 251-252, and Table 2, Fig. 8. Response:

Thanks for the suggestion. We have now included statistical test results in Table 2 and Figure 8. All trend values mentioned in that paragraph are statistically significant at the 95% confidence level. We have added this sentence to the manuscript.

86

Line 269-270: when the authors discuss the "largest contribution of East and 87 South Asia", does the authors mean East and South Asia contributes most at 88 this altitude compared with other regions, or this altitude is where East and 89 90 South Asia contributes most for their contributions at different altitudes? As a 91 matter of fact, I think these several paragraphs are terribly written (line 264-290). Keep in mind that, when you talk about the contribution, you are 92 comparing between different source regions as well as the altitudes. I highly 93 suggest the authors reorganize these several paragraphs. 94

95 **Response**:

We have now revised these paragraphs as follows to avoid the confusion: 96 97 "Aerosols are often transported across continents in the free troposphere rather than near the surface, resulting in a higher relative contribution of non-local 98 sources to the aerosol concentration at higher altitudes than near the surface. 99 100 Figure 6 shows the vertical profiles of absolute and relative contributions of major source regions to sulfate and BC concentrations in the Arctic. Different 101 source regions have very distinct vertical distributions of their contributions. 102 103 Below 1 km, Arctic local emissions account for the majority of Arctic sulfate and BC concentrations. For BC and sulfate located between 1 km and 5 km, 104 emissions from Russia are the major sources. Above 8 km, East Asia and South 105 Asia are the major source regions of the Arctic aerosol concentrations, which is 106 consistent with results using other models (e.g., Shindell et al., 2008). Arctic 107 and Russia have their maximum absolute contributions at 0.2 km and 1.4 km, 108 respectively. Europe and North America have their maximum absolute 109 contributions around 2 km. The contribution of East Asia and South Asia 110 increases with the increase of altitude, reaching their maximum contribution 111 values at 8 km and 11 km, respectively. 112

The changes in source contributions to the annual mean vertical profile of sulfate and BC concentrations over the Arctic between 2014–2018 and 1980– 1984 are shown in Fig. 7. Below 6 km, due to the effective emission reduction, the contribution from Europe and Russia to the Arctic sulfate was each decreased by nearly 0.1 μg m-3 in 2014–2018, compared to 1980–1984. North

America contribution also had a slight decline below 2 km. Between 10–15 km, 118 contributions from South Asia and East Asia increased at the upper troposphere, 119 which is consistent with the increase in emissions over these regions, leading 120 to a combined increase in sulfate concentration of up to 0.1 µg m-3 at the upper 121 troposphere of the Arctic. The BC concentration below 2 km contributed by 122 Arctic and Russia emissions each had a decrease of up to 2 ng m-3, which 123 dominated the decrease of BC concentration in the Arctic lower atmosphere. 124 Similar to sulfate, BC concentrations contributed by East Asia and South Asia 125 increased in the high altitudes, mainly due to increased emissions in these two 126 regions, offsetting the decrease in column burden owing to the reduced loading 127 in the lower atmosphere." 128 129 130 Editorial comments: Line 35: explain what "61%" is compared to. 131 Response: It is a comparison between 1980–1984 and 2014–2018. We have now clarified 132 it in the text. 133 134 Line 38: the snow/ice albedo effect from BC refers to local or other source 135 regions? 136 Response: 137 Here, the snow/ice albedo effect from BC refers to both local and other source 138 regions. We have followed the suggestion in main comments to reorganize the 139 abstract to avoid confusion as such. 140 141 142 Line 98: add from which year for the 2-3% changes. 143 Response: Following the suggestion, we have now revised the sentence to "Based on the 144 chemical transport model (GEOS-Chem) simulations, Breider et al. (2017) 145 146 found that annual sulfate and BC concentrations decreased by 2–3% per year over the Arctic during 1980-2010." 147 148 Line 122: change "observational" to "observation" 149 150 Response: 151 Following the suggestion, we have now revised the sentence to "Sulfate and BC concentrations from the CAM5-EAST model and observations at remote 152 Arctic stations are compared." 153 154 Line 153: EAST was already defined. 155 Response: 156 Deleted. 157 158 159 Line 181-182: Technically, neither Fig 1 nor Fig 2 showed the emission changes from "1980-2010" "from the 16 source regions". 160 Response: 161

Figure 1 shows the spatial distribution of annual mean SO2 and BC emissions 162 averaged over 1980-2018 from the 16 source regions and Figure 2 shows 163 time series of annual anthropogenic SO2 and BC emissions of major tagged 164 source regions and other regions (OTH, including ANT, CAM, CAS, MDE, 165 NAF, PAN, SAM, SEA, and SAF/NAM). In order to better see the time series 166 of annual emissions of other regions (OTH) individually, we have added the 167 time series these emissions in the supplementary materials (Fig. S1), which is 168 also shown below. 169 170

---

## Author Comment (AC3) · 26 May 2020

| 1        | Manuscript # acp-2020-3                                                                    |
|----------|--------------------------------------------------------------------------------------------|
| 2        | Posponsos to Poforoo #3                                                                    |
| 3
1   | Responses to Referee #5                                                                    |
| 4
5   | Review of "Source attribution of Arctic aerosols and associated Arctic warming             |
| 6        | trend during 1980-2018" by Ren et al.                                                      |
| 7        | This paper presents a modelling study of the impacts of changing SO 4 and BC    |
| 8        | on the Arctic atmospheric composition, radiative forcing, and temperature.                 |
| 9        | Modelled and measured SO 4 and BC are presented in the Arctic from 1980-        |
| 10       | 2018 at a handful of surface measurement sites. A tagged version of CAM5 is                |
| 11       | used to quantify the source contributions from different continental geographic            |
| 12       | regions to the Arctic BC and SO 4 concentrations both at the surface and in the |
| 13       | vertical column. The paper present interesting results that are important for              |
| 14       | understanding the rapidly warming Arctic. The authors conclude that about 20%              |
| 15       | of Arctic warming can be attributed to the combination of BC and SO4.                      |
| 16       | We thank the reviewer for all the insightful commente. Polow, places are our               |
| 10       | point-by-point response (in blue) to the specific comments and suggestions                 |
| 10       | and the changes that have been made to the manuscript in an effort to take                 |
| 20       | into account all the comments raised here                                                  |
| 21       |                                                                                            |
| 22       | I suggest only the following minor revisions below before publishing:                      |
| 23       | lines 130-131: is there a primary reference for CAM5 and CESM that you can                 |
| 24       | reference here?                                                                            |
| 25       | Response:                                                                                  |
| 26       | Thanks for the suggestion. We have now added the primary reference for                     |
| 27       | CESM as follows: "The global aerosol-climate model CAM5, which is the                      |
| 28       | atmospheric component of the earth system model CESM (Community Earth                      |
| 29       | System Model, Hurrell et al., 2013) developed at the National Center for                   |
| 30       | Atmospheric Research (NCAR), is used to simulate Arctic aerosols and climate               |
| 31       | for years 1980–2018 (after one-year model spin-up).                                        |
| 32
22 | lines 1/3-1/1: what is the source for the specified sea surface temperatures               |
| 33
34 | sea ice concentrations etc?                                                                |
| 35       | Response:                                                                                  |
| 36       | Sea surface temperatures and sea ice concentrations are created from the                   |
| 37       | merged Reynolds/HADISST products, as described in Hurrell et al. (2008).                   |
| 38       | Solar radiation and GHGs follow the CMIP6 configuration for AMIP-type of                   |
| 39       | simulations. We have now included these details in the manuscript.                         |
| 40       |                                                                                            |
| 41       | lines 209-210: was the modelled precipitation compared to measured                         |
| 42       | precipitation? Was wet deposition of model validated against measurements?                 |
| 43       | Response:                                                                                  |
| 44       | The performance of CAM5 in aerosol wet deposition and transport to the Arctic              |

has been specifically evaluated and improved in previous studies (e.g., Liu et 45 al., 2011; Wang et al., 2013; Qian et al., 2014; Yang et al., 2018a. To address 46 this comment and follow a suggestion from one of the other reviewers, we have 47 revised the sentence to "According to previous CAM5 studies on aerosol wet 48 removal and long-range transport, the model underestimates aerosol 49 concentrations in spring, likely due to biases in parameterizations of convective 50 transport and wet scavenging of aerosols (Bond et al., 2013, Liu et al., 2011, 51 Wang et al., 2013; Qian et al., 2014; Yang et al., 2018a)." 52

53

Fig 5/line 241: it needs to be clarified that Fig 5 is the model average in the Arctic (>66.5  $^{\circ}$ N).

56 **Response**:

57 Following the suggestion, we have now revised the sentence to "The absolute 58 and relative source contributions of emissions from the major source regions to 59 the simulated annual mean near-surface sulfate and BC concentrations 60 averaged over the Arctic (66.5°N–90°N) are shown in Fig. 5."

61

line 252: was that rise in BC seen in the observations? e.g., consistent with BC
 seen at Alert?

64 **Response**:

Yes, we have now revised the sentence to "Simulated Arctic BC concentration also shows a considerable decline before 2000, but a slight rise after 2000, which is consistent with the BC observations at Alert."

68

69 line 263: "in the Arctic" ... and Russia?

70 Response:

Yes, we have now revised the sentence to "To further reduce present-day or future aerosols in the Arctic, efforts can be made to control local sources in the

- 73 Arctic as well as emissions from Russia."
- 74

line 316: is the effect of BC deposition on snow/reduction of albedo included in
 this? I think not because that effect is discussed later, but could clarify here that
 this value is just for atmospheric BC effect.

78 Response:

No, the effect of BC deposition on snow/reduction of albedo is not included in
it. This value is for atmospheric BC effect only. We have now revised the text
to "The Arctic sulfate exerts a negative RFari primarily by scattering incoming
solar radiation back into the space, with the forcing in a range of -0.4~0 Wm-2.
The atmospheric BC can absorb solar radiation in the atmosphere and leads to
a positive RFari of 0.1~0.4 Wm-2 in the Arctic."

85

Section 5/line 400: Can you add some discussion as to how the model bias
 affects your conclusions? E.g. would your estimates of SO4 and BC
 temperature impacts be greater or lesser if the model were corrected to

89 accurately reflect the measurements?

90 **Response**:

91 Thanks for the suggestion. We have now revised the sentence to "Considering

92 that the model underestimates the magnitude of sulfate and BC concentrations,

the estimated impact on Arctic temperature from sulfate and BC could be even

94 larger if the model were able to accurately reproduce the measurements in the95 Arctic."

96

Data availability: please add where the Arctic BC & SO4 measurements can be
 found in this section (e.g., EBAS database link).

- 99 Response:
- 100 **Added**.
- 101

Figs 1-2, and 5-7: please make sure the regional colours are consistent in all of these plots. e.g., colour X for RBU, colour Y for EUR, etc, in all 5 figures the same.

105 **Response**:

106 We have now made the regional colors consistent in all plots.

107

Fig 3 (4): Clarify in the caption that the black is from measurements, and the blue and green are modelled. E.g., "Measured seasonal means are denoted by...". "Stacked contours represent the modelled Arctic..."

111 Response:

112 Thanks for the suggestion. We have now revised the figure caption to:

Figure 3. Surface concentrations of sulfate aerosols (µg m-3) in spring (March– 113 May) and summer (June-August) at four locations (Alert, Station Nord, Ny-114 Alesund, Kevo) in the Arctic during 1980–2018. Seasonal means are denoted 115 116 by solid black circles, medians as short horizontal bars, and the 25th to 75th percentile ranges as vertical bars. Stacked colors represent modeled 117 contributions from the Arctic (blue) and non-Arctic anthropogenic source region 118 (green). The observations denoted by solid black circles are obtained from 119 European Monitoring and Evaluation Programme and World Data Centre for 120 Aerosols database (http://ebas.nilu.no) and Breider et al. (2017). Black 121 122 triangles at Ny-Alesund for the period 1980–1981 show mean observations from Heintzenberg and Larssen (1983). Black diamond at Ny-Alesund in 123 summer shows median non-sea-salt sulfate concentration from Maenhaut et al. 124 (1989). Open circles in the spring for Ny-Ålesund are March–April mean values 125 (Sirois and Barrie, 1999). Note that the vertical coordinates use logarithmic 126 scales. 127

128

Fig 3: why is Barrow not shown? Fig 4: why is St Nord not shown? Fig 5: specify that this is the Arctic (>66.5 °N) average. As mentioned above, use the same regional colour scheme here as in Fig 1(a) & Fig 2. Fig 6 & 7: match the regional colours to Fig 5. 133 **Response:**

134 The data of Barrow and St Nord sites are relatively scarce. We only selected135 sites with more than 20 observation samples.

Following the suggestion, the caption Figure 5 has been revised to "Time series (1980–2018) of absolute (left,  $\mu$ g m-3) and relative (right, %) contributions of emissions from the major source regions to the simulated annual mean nearsurface sulfate and BC concentrations averaged over the Arctic (66.5°N–90°N).

- Fig 2, Fig 5, Fig 6 and Fig 7 have now been revised to use the same regionalcolor scheme.
- 143
- 144
- 145 **Reference**:

Breider, T. J., Mickley, L. J., Jacob, D. J., Ge, C., Wang, J., Sulprizio Payer, M.,
Croft, B., Ridley, D. A., McConnell, J. R., Sharma, S., Husain, L., Dutkiewicz,
V. A., Eleftheriadis, K., Skov, H., and Hopke, P. K.: Multidecadal trends in
aerosol radiative forcing over the Arctic: Contribution of changes in
anthropogenic aerosol to Arctic warming since 1980, J. Geophys. Res. Atmos.,
122, 3573–3594, https://doi.org/10.1002/2016JD025321, 2017.

152

Bond, T. C., Doherty, S. J., Fahey, D. W., Forster, P. M., Berntsen, T., 153 DeAngelo, B. J., Flanner, M. G., Ghan, S., Kärcher, B., Koch, D., Kinne, S., 154 Kondo, Y., Quinn, P. K., Sarofim, M. C., Schultz, M. G., Schulz, M., 155 Venkataraman, C., Zhang, H., Zhang, S., Bellouin, N., Guttikunda, S. K., Hopke, 156 P. K., Jacobson, M. Z., Kaiser, J. W., Klimont, Z., Lohmann, U., Schwarz, J. P., 157 Shindell, D., Storelvmo, T., Warren, S. G., and Zender, C. S.: Bounding the role 158 of black carbon in the climate system: A scientific assessment, J. Geophys. 159 Res.-Atmos., 118, 5380–5552, https://doi.org/10.1002/jard.50171, 2013. 160 161

Heintzenberg, J., Larssen, S.: SO2 and SO4 = in the Arctic: Interpretation of
observations at three Norwegian Arctic-Subarctic stations, Tellus B, 35B(4),
255–265, https://doi.org/10.1111/j.1600-0889.1983.tb00028.x, 1983.

165

Hurrell, J. W., Holland, M. M., Gent, P. R., Ghan, S., Kay, J. E., Kushner, P. J., 166 Lamarque, J. F., Large, W. G., Lawrence, D., Lind- say, K., Lipscomb, W. H., 167 Long, M. C., Mahowald, N., Marsh, D. R., Neale, R. B., Rasch, P., Vavrus, S., 168 Vertenstein, M., Bader, D., Collins, W. D., Hack, J. J., Kiehl, J., and Marshall, 169 S.: The Community Earth System Model A Framework for Collaborative 170 Research, Β. Am. Meteorol. Soc., 94, 1339-1360, 171 https://doi.org/10.1175/BAMS-D-12-00121.1, 2013. 172

173

Hurrell, J.W., J.J. Hack, D. Shea, J.M. Caron, and J. Rosinski,: A New Sea
Surface Temperature and Sea Ice Boundary Dataset for the Community
Atmosphere Model. J. Climate, 21, 5145–5153,

- 177 https://doi.org/10.1175/2008JCLI2292.1, 2008.
- 178

182

Liu, J., Fan, S., Horowitz, L.W., and Levy II, H.: Evaluation of factors controlling
long-range transport of black carbon to the Arctic, J. Geophys. Res., 116,
D04307, https://doi.org/10.1029/2010JD015145, 2011.

- Maenhaut, W., Cornille, P., Pacyna, J. M., & Vitols, V.: Arctic air chemistry trace element composition and origin of the atmospheric aerosol in the Norwegian Arctic, Atmos. Environ., 23(11), 2551–2569, https://doi.org/10.1016/0004-6981(89)90266-7, 1989.
- 187

Qian, Y., Wang, H., Zhang, R., Flanner, M. G., Rasch, P. J.: A sensitivity study
on modeling black carbon in snow and its radiative forcing over the Arctic and
Northern China. Environ. Res. Lett., 9,064001, https://doi.org/10.1088/1748-
9326/9/6/064001, 2014.

192

Sirois, A., & Barrie, L. A.: Barrie (Arctic lower tropospheric aerosol trends and
composition at Alert, Canada: 1980–1995, J. Geophys. Res., 104(D9), 11,599–
11,618, https://doi.org/10.1029/1999JD900077, 1999.

196

Wang, H., Easter, R. C., Rasch, P. J., Wang, M., Liu, X., Ghan, S. J., Qian, Y.,
Yoon, J.-H., Ma, P.-L., and Vinoj, V.: Sensitivity of remote aerosol distributions
to representation of cloud–aerosol interactions in a global climate model,
Geosci. Model Dev., 6, 765–782, https://doi.org/10.5194/gmd-6-765-2013,
2013.

202

---

## Referee Report (RR1)

Review of "Source attribution of Arctic black carbon and sulfate aerosols and associated Arctic surface warming during 1980–2018" by Ren et al.

This is my second review of this paper, which presents a modelling study of the impacts of changing SO4 and BC on the Arctic atmospheric composition, radiative forcing, and temperature. Modelled and measured SO4 and BC are presented in the Arctic from 1980-2018 at a handful of surface measurement sites. A tagged version of CAM5 is used to quantify the source contributions from different continental geographic regions to the Arctic BC and SO4 concentrations both at the surface and in the vertical column. The paper present interesting results that are important for understanding the rapidly warming Arctic. The authors addressed all reviewers comments thoughtfully and thoroughly. I have no further suggestions for improvement, and recommend that the revised manuscript be published as is.